# An extended interaction site determines binding between AP180 and AP2 in clathrin mediated endocytosis

Samuel Naudi-Fabra[1,2], Carlos A. Elena-Real[1], Ida Marie Vedel [1], Maud Tengo[2], Kathrin Motzny[1], Pin-Lian Jiang[1], Peter Schmieder[1], Fan Liu [1] & Sigrid Milles [1,2] ✉

The early phases of clathrin mediated endocytosis are organized through a highly complex interaction network mediated by clathrin associated sorting proteins (CLASPs) that comprise long intrinsically disordered regions (IDRs). AP180 is a CLASP exclusively expressed in neurons and comprises a long IDR of around 600 residues, whose function remains partially elusive. Using NMR spectroscopy, we discovered an extended and strong interaction site within AP180 with the major adaptor protein AP2, and describe its binding dynamics at atomic resolution. We find that the 70 residue-long site determines the overall interaction between AP180 and AP2 in a dynamic equilibrium between its bound and unbound states, while weaker binding sites contribute to the overall affinity at much higher concentrations of AP2. Our data suggest that this particular interaction site might play a central role in recruitment of adaptors to the clathrin coated pit, whereas more transient and promiscuous interactions allow reshaping of the interaction network until cargo uptake inside a coated vesicle.

Clathrin Mediated Endocytosis (CME) is the most widespread form of endocytosis[1], leading to the formation of small vesicles (from around 20 to 60 nanometers in diameter[2]) that transport cargo proteins into the cytoplasm. It is key to many cellular processes such as nutrient absorption, membrane composition regulation, signaling, adhesion, and viral entryway into the cell. A lattice formed by a number of clathrin triskelia builds the exoskeleton of the clathrin-coated vesicles. Clathrin itself, however, cannot bind to the membrane or the incorporated cargo and requires a complex network of adapter and accessory proteins to enable controlled cargo uptake[3]. The major adapter complex AP2, composed of four different subunits, recognizes cargo, as well as the signaling lipid phosphatidylinositol-(4,5)-bisphosphate (PIP2), and is recognized by clathrin. Its α and β2 subunits, the two largest subunits, form the core of the AP2 complex and comprise two intrinsically disordered linkers followed by C-terminal appendage domains[4]. The appendage domains and linkers are known to bind

clathrin and possess binding pockets for accessory proteins, also called clathrin associated sorting proteins (CLASPs)[5,6]. CLASPs are usually composed of an N-terminal folded domain, which binds the membrane and can often recognize certain cargoes, and a large C-terminal intrinsically disordered domain, comprising small linear motifs (SLiMs) that interact with the AP2 appendage domain, the clathrin terminal domain, or folded domains of other CLASPs[7,8]. Amongst those count, for example, proteins that take part in the pioneering module, such as the F-BAR domain-only protein 1 and 2 (FCHO1/2), the clathrin assembly lymphoid myeloid leukemia (CALM) protein, or the Epidermal growth factor substrate 15 (Eps15)[9].

One such CLASP is the 91 kDa monomeric clathrin Assembly Protein AP180, the synaptic counterpart of the ubiquitous CALM protein. Both proteins possess a very similar N-terminal folded domain (termed AP180 N-terminal homology - ANTH - domain) that binds the membrane through PIP2[10] and seems to be responsible for sorting

[1]Leibniz-Forschungsinstitut für Molekulare Pharmakologie, Robert-Rössle-Straße 10, 13125 Berlin, Germany. [2]Université Grenoble Alpes, CNRS, CEA, IBS, F-38000 Grenoble, France. ✉e-mail: milles@fmp-berlin.de

soluble N-ethylmaleimide-sensitive factor attachment protein receptors (SNAREs), by directly binding to synaptobrevin[11–13]. While the C-terminal intrinsically disordered region (IDR) of CALM is 300 residues in length, AP180 harbors an IDR of impressive 600 residues in length that possesses a high number of the small clathrin binding motif Asp-Leu-Leu/Phe (DLL/DLF). The sheer number of the low affinity interactions is supposed to increase overall affinity towards clathrin and may contribute to bridging different clathrin triskelia together[14,15]. Similar to AP2, AP180 has been shown to promote clathrin assembly and, together, AP2 and AP180 seem to develop synergistic effects to dramatically enhance the formation of clathrin lattices; a finding that has led to the identification of a putative AP2 binding region through biochemical pull-down experiments of truncated AP180[16]. Indeed, pull-down assays of CLASPs and their different truncation constructs along with X-ray crystallography of small CLASP-derived peptides and their folded partner proteins have been the main driving forces towards a mechanistic understanding of CLASPs' function[6,17,18].

In order to understand the different putative binding sites in the context of their native sequence, in this study we set out to study the approximately 600 residue long AP180 IDR in its entirety. We use nuclear magnetic resonance (NMR) spectroscopy to provide a comprehensive molecular analysis of AP180's IDR and assess its conformational sampling, backbone dynamics, and interaction landscape with the clathrin heavy chain terminal domain and with the two AP2 appendage domains α and β2. The protein behaves like an IDR with very little sampling of secondary structure, such as α-helices or extended structures, and binds its interaction partners through various small and hydrophobic binding sites at a high level of promiscuity

and with affinities in the hundreds of micromolar range. Unexpectedly, we discovered an interaction site between AP180 and the AP2 β2-appendage domain that had not been previously identified and that leads to an interaction orders of magnitude stronger than the other interaction sites within AP180, regardless of the interaction partner (AP2 appendage domains or clathrin heavy chain terminal domain). The interaction site, spreading over an extended sequence stretch in AP180, demonstrates concerted binding behavior with an on-off binding rate of around 600 s⁻¹. Based on the interplay between this high affinity interaction site and other low affinity binding sites, we propose distinct functions for these interactions for recruitment and rearrangement of partners at the nascent and growing endocytic pit, respectively.

## Results

### The IDR of AP180 behaves like a prototypical disordered chain

We first analyzed the domain architecture of the human AP180 (Fig. 1A), which comprises an around 300 residue-long ANTH domain, followed by a domain predicted to be intrinsically disordered in its entirety[19,20], and contains sequence motifs for the interaction with the clathrin heavy chain terminal domain (DLL/DLF), or the AP2 α and β2 appendage domains (DPF and FxDxF)[21].

In order to study the large, 61 kDa, AP180 IDR (residues 281–898) and its interactions at molecular resolution using NMR spectroscopy, we followed a divide and conquer approach: we designed different overlapping segments of the AP180 IDR of around 200 residues in length each (Supplementary Fig. 1). This strategy allowed us to simplify the NMR spectra as compared to the full length IDR (AP180$_{IDR}$) and to

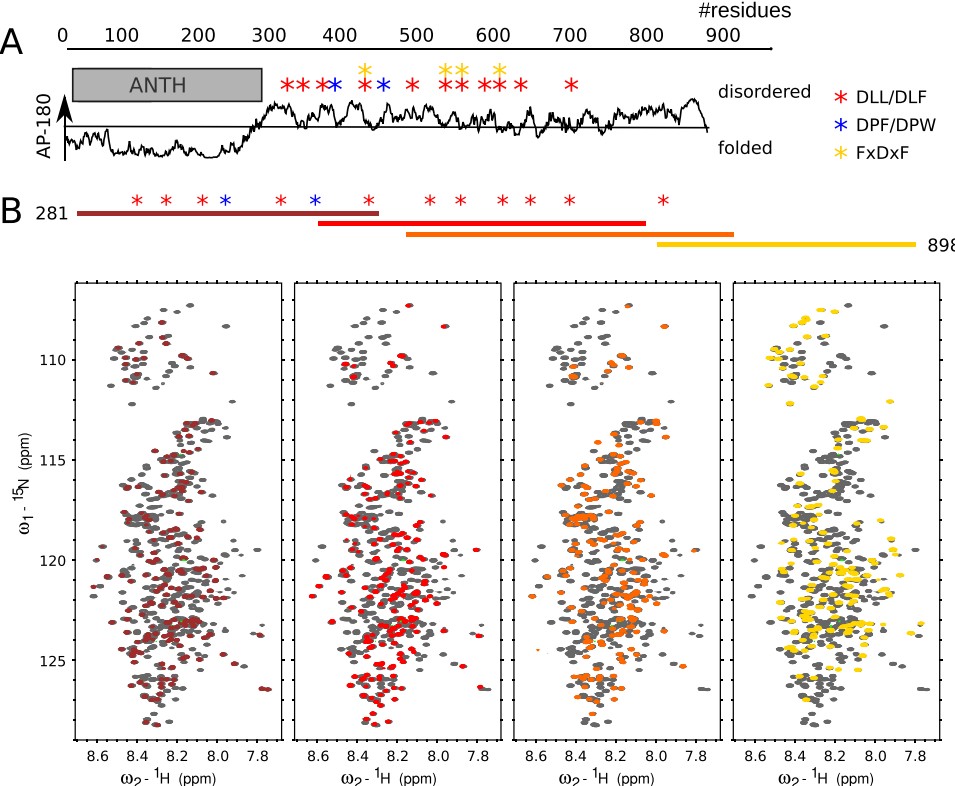

**Fig. 1 | Scheme of AP180 and ¹H-¹⁵N HSQC spectra of its different constructs. A** Disorder prediction (IUPred[19]) of AP180 along its sequence. The gray box represents the folded ANTH (AP180 N-terminal homology) domain. Stars above the disorder prediction represent putative interaction sites with the clathrin heavy chain terminal domain and with the AP2 appendage domains as indicated in the figure. **B** ¹H-¹⁵N heteronuclear single quantum coherence (HSQC) spectra of AP180 IDR and its different segments. IDR contains residues 281–898 of the full length

AP180. Smaller segments are illustrated in colors (281–500, 471–700, 540–740, 720–898). Colors of the HSQC spectra match those in the illustration on top. The spectra of the individual segments are overlaid with the spectrum of AP180 IDR (in gray) and demonstrate good superposition. Stars indicate putative interaction sites based on a sequence analysis: red: DLL/DLF (clathrin binding), blue: DPF (AP2 binding), yellow: FxDxF (AP2 binding).

ease the protein production. AP180$_{IDR}$ itself was purified under denaturing conditions and transferred into native buffer in the final purification step, allowing us to obtain sufficient quantity for NMR experiments.

We recorded $^1$H-$^{15}$N heteronuclear single quantum coherence (HSQC) spectra of the different AP180 segments and superimposed them onto the spectrum of AP180$_{IDR}$ (Fig. 1). A very good overlap between the spectra of the individual segments and AP180$_{IDR}$ testifies to the validity of this approach and suggests that specific long-range interactions within AP180$_{IDR}$ are largely absent. We thus assigned the backbone resonances of the individual segments (Supplementary Figs. 2–6) and transferred the assignment onto the $^1$H-$^{15}$N HSQC spectrum of AP180$_{IDR}$. $^{13}$C (C$_\alpha$, C$_\beta$, CO) secondary chemical shifts (SCSs) of the individual AP180 segments were calculated by subtracting the acquired chemical shifts from those expected from random coil[22]. With SCS values around zero, AP180 behaves like a prototypical disordered protein across the entire length of its IDR, where secondary structural elements are only sparsely populated (Supplementary Figs. 7, 8). The very C-terminal stretch of the protein (AP180$_{720-898}$), which is devoid of classical putative binding motifs, stands out with a helical element that is transiently populated between residues 750 and 770 followed by a long stretch with elongated structural sampling nearly until the C-terminus of the protein. This is particularly evident from the secondary structure propensities (SSPs)[23], which are derived from carbon SCSs and take up values between −1 (fully elongated/β-strand structure) and 1 (fully formed helix). This region is also characterized by overall higher $^{15}$N transverse relaxation rates (Supplementary Fig. 8) compared to the rest of the IDR. Even though slightly lower throughout the entire protein, probably due to the motional drag that the long IDR exerts, $^{15}$N R$_{1\rho}$ rates of the different segments follow those measured in the full AP180$_{IDR}$ very closely and are reminiscent of rates expected for a disordered chain (Supplementary Fig. 8), suggesting similar characteristics of chain motion between the different segments and AP180$_{IDR}$.

## CHC$_{TD}$ binds to all AP180 DLL/DLF motifs with low affinities

The sequence of AP180's IDR harbors many DLL and DLF motifs (Fig. 1, Supplementary Fig. 1), expected to bind clathrin and assist clathrin assembly[14,24]. In order to gain insight into the binding of all those motifs to clathrin when in the context of their native sequence environment, we titrated increasing amounts of the clathrin heavy chain terminal domain (CHC$_{TD}$, residues 1–363 of the full length protein) into the different AP180 segments as well as into AP180$_{IDR}$. Around the putative interaction sites within the sequence of AP180, we observe local chemical shift changes and peak broadening as the concentration of CHC$_{TD}$ increases (Fig. 2A). This peak broadening is also manifested as increased $^{15}$N R$_{1\rho}$ relaxation rates around the interaction sites (Fig. 2, Supplementary Fig. 9). Intriguingly, not only the classical clathrin binding motifs DLL or DLF contribute to binding, but also motifs normally associated with binding AP2, such as DPF or FxDxF, albeit with a smaller increase in $^{15}$N R$_{1\rho}$ rates in the course of the titration. Indeed, hydrophobic residues seem to engage in binding in a very general way, independently of their association in one of the known endocytic binding motifs. In order to analyze any potentially unknown sequence specificity, we selected sequence stretches of 21 residues in length around the highest $^{15}$N R$_{1\rho}$ in AP180$_{IDR}$, and centered them around a hydrophobic residue. This alignment validates that, despite the fact that DLL and DLF motifs are prominent among the binding sequences, other sequences are also prone to bind. A slight preference for a glycine or alanine following L or F seems to exist, while no other flanking sequences are enriched in the alignment (Fig. 2D).

While peak broadening in the course of the titration could be indicative of intermediate exchange in the microsecond to millisecond regime, CPMG (Carr-Purcell-Meiboom-Gill) relaxation dispersion curves did not show presence of intermediate exchange (flat curves,

Supplementary Fig. 10). Instead, the peak broadening occurs due to slowed rotational tumbling of the bound IDR stretch interacting in fast exchange with the relatively large and folded partner protein. In this case, transverse relaxation of AP180 will be determined by the weighted average of the rates in the bound and unbound states. The relaxation rates are thus a function of the fast motion that AP180 exerts in its unbound state, leading to a contribution of low R$_{1\rho}$ values, and the slow rotational tumbling when bound to the comparatively large and folded CHC$_{TD}$, leading to a contribution of high R$_{1\rho}$ values. A higher rate would thus be indicative of a stronger affinity. We therefore sorted the selected sequence stretches surrounding the main interaction sites by height of the central $^{15}$N R$_{1\rho}$ value and re-assessed the sequence properties of the interaction sites, showing that DLL motifs end up higher in the list than DLF motifs on average (Supplementary Table 1).

The areas with increased $^{15}$N R$_{1\rho}$ rates within the different AP180 segments are rather well separated. At the different CHC$_{TD}$ concentrations, they increase only locally and these regions of increased R$_{1\rho}$ rates are separated by stretches with rates reminiscent of the unbound state, suggesting that the different binding sites bind independently from each other. To a large extent, this is even true for the cluster of DLL/DLF motifs around residues 550–730 of AP180, which encouraged us to try and extract residue-wise equilibrium dissociation constants (K$_D$) from the residues with the highest $^{15}$N R$_{1\rho}$ rates around one interaction site[25]. Under the assumption of weak binding, a linear fit of the rates versus percentage of CHC$_{TD}$ binding allowed us to approximate residue-wise K$_D$ values on the order of hundreds of micromolar to several millimolar (Fig. 2E, Supplementary Table 2), in agreement with previous studies on AP180 binding to clathrin as well as binding of other SLiMs to their folded binding partners[24,25].

No notable differences between the different AP180 segments and the full AP180$_{IDR}$ constructs are apparent, except for overall slightly lower $^{15}$N R$_{1\rho}$ rates for AP180$_{IDR}$ compared to the segments at a specific CHC$_{TD}$ concentration. This might stem from the presence of other competitive interactions along the chain that are not contained in the individual segments. The C-terminal stretch of AP180, which is devoid of clathrin binding motifs (AP180$_{720-898}$) does not show evidence of interaction with CHC$_{TD}$. Mildly increased $^{15}$N R$_{1\rho}$ rates of the C-terminus in AP180$_{IDR}$ in the presence of CHC$_{TD}$ may stem from increased tumbling times due to CHC$_{TD}$ binding elsewhere in the disordered chain.

## AP2α binds to AP180 in a promiscuous way with affinities similar to CHC$_{TD}$

While AP180 is recognized for enhancing clathrin assembly, synergy with AP2 for its assembly function has been proposed[16] and the α-appendage domain of AP2 has been shown to pull down AP180[26]. We have thus analyzed the binding between AP180 and the α-appendage domain of AP2 (from now on called AP2α). We employed a similar approach to our investigation of the binding of AP180 and CHC$_{TD}$, and titrated increasing concentrations of AP2α into AP180$_{IDR}$ and the different AP180 segments. Even though only two putative binding sites for AP2α are contained in the entire length of the AP180 IDR (around residues 400 and 480), we observe interactions, accompanied by increased $^{15}$N R$_{1\rho}$ relaxation rates, at various positions along the chain of AP180 (Fig. 3, Supplementary Fig. 11). Binding between AP180 and AP2α thus seems to be remarkably promiscuous, with classical clathrin binding motifs DLL/DLF binding with apparently equal strength as DPF motifs or FxDxF, normally known to bind the AP2 α- and β2 appendage domains. A thorough analysis of AP180 $^{15}$N R$_{1\rho}$ rates reveals that not only DLL/DLF, DPF and FxDxF motifs engage in binding, but that AP2α seems to interact with small hydrophobic clusters within the sequence of AP180 in a very general sense (Supplementary Fig. 11). Intriguingly, the very C-terminal stretch of AP180, encompassed in the AP180$_{720-898}$ segment, is devoid of known binding motifs but contains hydrophobic residues, and yet does not show any sign of interaction with AP2α. In

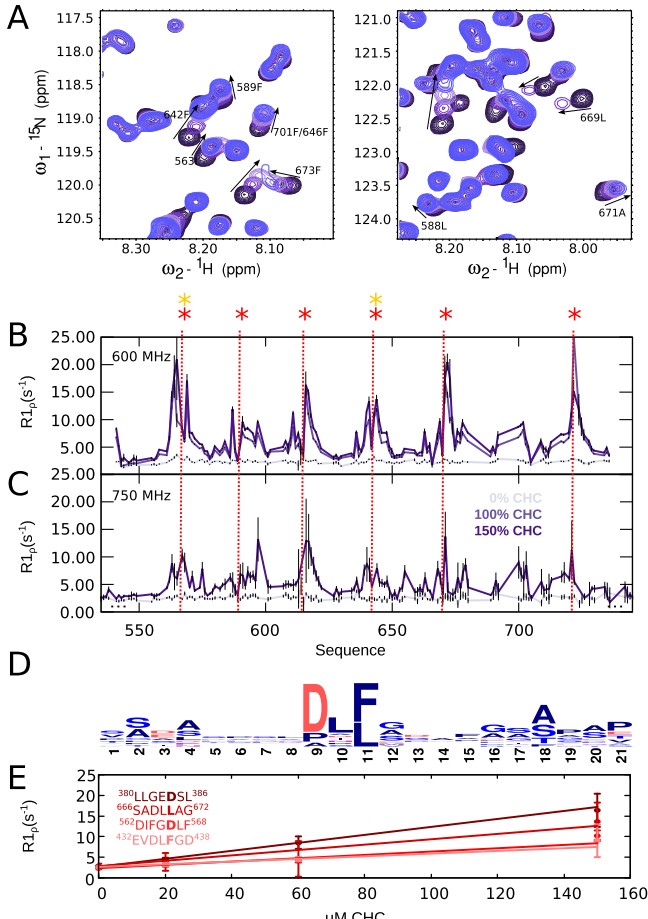

**Fig. 2 | Interaction of AP180 with CHC_TD. A** Zooms into the spectra of AP180_540-740 in the absence and the presence of increasing concentrations of CHC_TD (0%, 50%, 100%, 150%, from dark to light blue). **B** $R_{1\rho}$ relaxation rates of AP180_540-740 and **C** of AP180_IDR (only the region of 540–740 is plotted) in the absence and the presence of increasing concentrations of CHC_TD. Colors are as indicated in the figure. Interaction motifs are indicated as stars and dashed lines, color code as in Fig.1. **D** Sequence conservation around the major interaction sites. Regions with increased $R_{1\rho}$ relaxation rates upon addition of CHC_TD within AP180_IDR were selected and centered around the hydrophobic residues, which usually showed the highest relaxation rates. The size of the amino acid (one letter code) illustrates the abundance within the total 13 interaction regions (see Supplementary Table S1 for the full list). The image was generated using WebLogo (http://weblogo.berkeley.edu/logo.cgi). **E** $R_{1\rho}$ relaxation rates of AP180_IDR plotted against the concentration of CHC_TD and fitted with a binding curve (see materials and methods). The fitted residue is shown in bold in the figure. $K_D$ values are 177 µM, 294 µM, 549 µM, and 700 µM, respectively. A detailed list of all affinities can be found in Supplementary Table 2. Errors of relaxation rates were derived from the experimental uncertainty. Source data are provided as a Source Data file.

order to assess the binding strengths of AP2α to the individual motifs within AP180, we undertook an analysis of the measured relaxation rates of AP180 at different admixtures of AP2α, which revealed equilibrium dissociation constants that are overall comparable to those observed for CHC_TD binding (hundreds of micromolar).

**AP2β2 binds to a special site on AP180 at unexpected length and strength**

To complete the interaction landscape of AP180 with known endocytic partners, we then assessed binding to the AP2 β2-appendage domain (from now on called AP2β2), which, as AP2α, has been pulled down by various constructs of AP180[16]. We employed the same strategy as for binding to CHC_TD and AP2α to assess binding to AP2β2 and titrated

increasing amounts of AP2β2 into different $^{15}$N labeled AP180 segments as well as the full AP180_IDR. In contrast to experiments performed with CHC_TD or AP2α as binding partners, we identified a distinct stretch on AP180 that possessed striking binding characteristics: With as little as 5% added AP2β2, many peaks in the region between residue 430 to 500 nearly disappear and demonstrate increased $^{15}$N $R_{1\rho}$ relaxation rates (Fig. 4), arguing for a higher affinity interaction as compared to canonical SLiM interactions. The sequence length of the AP180 region involved in binding is also distinct with respect to known interaction motifs, it encompasses two spatially close sites, only a few residues apart: The first spans residues 435–445 and harbors a combination of clathrin binding (DLF) and AP2β2 binding (FxDxF) motifs (DLFGDAF). The second spans residues 470–500, and comprises an AP2α binding (DPF) and a clathrin binding (DLF) motif. In contrast to the SLiMs normally known to mediate endocytic interactions between CLASPs, clathrin and the major adapter complex AP2, which are defined to have an overall length between around 3 and 15 residues[27], we term this interaction site involved in AP180 binding to AP2β2 an 'extended interaction motif' (EIM). The region comprising the EIM with AP2β2 is devoid of significantly populated secondary structure elements, similar to the rest of the AP180 IDR.

Similar to binding of AP180 to CHC_TD or AP2α, regions outside the EIM show features reminiscent of SLiM binding and a high level of promiscuity: all clathrin binding motifs within the segment AP180_540-740 bind to AP2β2 and lead to similar AP180 $^{15}$N $R_{1\rho}$ rates as those observed upon AP2α binding, thus arguing for similar affinities on the order of hundreds to thousands of micromolar. Addition of the same amount (50%) of AP2β2 to the full AP180_IDR, however, does not lead to $^{15}$N $R_{1\rho}$ rates nearly as high as those observed within AP180_540-740 upon interaction with DLL/DLF motifs. The reason for this is likely the presence of the EIM, with apparently much higher affinity towards AP2β2 that competes with binding to DLL/DLF motifs within the remainder of AP180_IDR (Supplementary Fig. 12). It is worth noting that at even higher AP2β2 concentrations (e.g. 500% compared to AP180), when EIM sites on all proteins are saturated, AP2β2 also engages in interactions with SLiMs, such as DLL/DLF motifs (Supplementary Fig. 13). Even though the EIM seems to determine the overall interaction between AP180 and AP2β2 (Fig. 5, Supplementary Fig. 12), the interaction landscape between AP180 and AP2β2 is context-dependent and determined by the relative concentrations of each partner.

While we can estimate the affinities between AP2β2 and the different short binding motifs to be similar to those measured for interactions between CHC_TD or AP2α with AP180, the – likely higher – affinity between the EIM and AP2β2 remained to be determined. Isothermal titration calorimetry (ITC) experiments of AP180_IDR with AP2β2 yielded a $K_D$ value of 6.97 ± 2.8 µM (mean ± standard deviation over three independent measurements) between the two proteins, while the average $K_D$ of AP180_399-598 with AP2β2 (over two independent measurements) was determined to be 5.1 ± 0.42 µM, thus extremely similar to the overall affinity between AP180_IDR and AP2β2 (Fig. 5). Based on our NMR results, we delineated the interaction site around the EIM even more precisely and conceived a construct that would comprise only the EIM, reaching from residues 430–500 of AP180. Its affinity of around 2.5 µM demonstrates that the EIM determines the overall affinity between AP180 and AP2β2, in agreement with the visual inspection of the $^{15}$N $R_{1\rho}$ rates of AP180_IDR in the presence of increasing concentrations of AP2β2 (Supplementary Fig. 12B). Notably, this affinity is orders of magnitude higher than affinities between known SLiMs contained in AP180 and their interaction partners (Supplementary Tables 2 and 3)[24].

**Platform and sandwich domains of AP2β2 interact with AP180_430-500**

The length of the EIM, as well as the comparably high affinity of this interaction site poses the question as to how it would bind to the

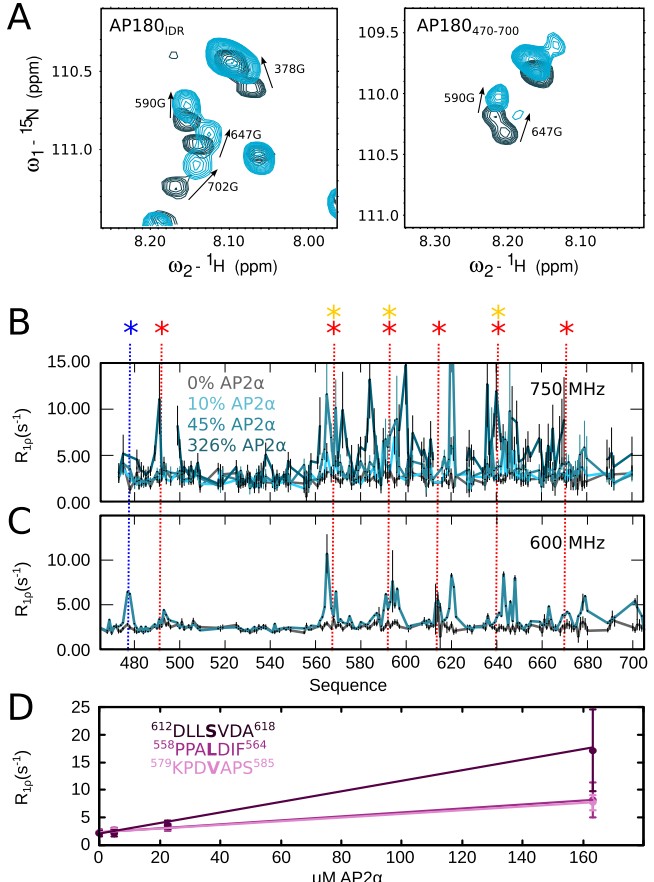

**Fig. 3 | Interaction of AP180 with AP2α. A** Zoom into the spectra of AP180$_{471-700}$ (right) and AP180$_{IDR}$ (left) in the absence (dark blue) and the presence (turquoise) of 45% AP2α at a concentration of 50 μM and 90 μM AP180, respectively. **B** R$_{1ρ}$ relaxation rates of AP180$_{471-700}$ and **C** of AP180$_{IDR}$ (only the region from 471 to 700 is plotted) in the absence and the presence of increasing concentrations of AP2α (10%, 45% and 326% at a concentration of 50 μM AP180 in **B**; 45% at a concentration of 90 μM in **C**). Colors are as indicated in the figure. Interaction motifs are indicated as stars and dashed lines, color code as in Fig.1. **D** R$_{1ρ}$ relaxation rates of AP180$_{471-700}$ plotted against the concentration of AP2α and fitted with a binding curve. K$_D$ values are 144 μM, 380 μM, and 414 μM, respectively. A detailed list of all affinities can be found in Supplementary Table 3. Errors of relaxation rates were derived from the experimental uncertainty. Source data are provided as a Source Data file.

surface of AP2β2. Both AP2α and AP2β2 are known to harbor binding sites for SLiMs contained in CLASPs, which have been mapped using crystallography and pull-down experiments[5,6,17,28,29]. Two such sites exist per appendage domain, one in the platform domain (C-terminal part of the appendage domain) and one in the sandwich domain (N-terminal part of the appendage domain) for AP2α as well as AP2β2, both able to independently bind the described linear motifs. Mutants of AP2β2 have been described to either alter (K842E, Y888E) or totally disrupt (Y815A) pull-down of AP180 by the GST-tagged AP2 β2-appendage domain[6]. K842 and Y888 reside on the C-terminal platform domain and Y815 on the N-terminal sandwich domain of the β2-appendage domain.

We assessed binding of AP180$_{399-598}$ to the AP2β2 mutants described (K842E, Y888E, Y815E) and measured $^{15}$N spin relaxation (R$_{1ρ}$) of AP180$_{399-598}$ at 50% and 100% excess of wild type and mutant AP2β2. Strikingly, binding of AP180$_{399-598}$ to AP2β2 persists for all mutants. This becomes obvious when focusing on the peaks within the EIM, which totally disappear at an excess of 50% wild-type AP2β2. In the presence of 50% AP2β2 mutants the respective peaks significantly

broaden (and show increased $^{15}$N R$_{1ρ}$ rates), albeit not to disappearance, demonstrating that the interaction persists, but is weakened (Fig. 6). This suggests that both platform and sandwich domains may be simultaneously bound by AP180's EIM, as none of the mutations abolishes, but all affect binding. Interestingly, the two DLF/FxDxF motifs towards the C-terminus of AP180$_{399-598}$ (DIFGDLFD, 562–569/DLFGTDAF, 587–594) seem to bind more strongly to all AP2β2 mutants (and Y815A in particular) than to the wild type AP2β2, showcased by significantly increased $^{15}$N R$_{1ρ}$ rates in the presence of AP2β2 mutants (Fig. 6). This observation is in agreement with our hypothesis that the EIM may contact both binding sites in the platform and sandwich domains of AP2β2 simultaneously. Weakening the overall interaction of the EIM through single point mutations might allow the much smaller and normally much more weakly binding SLiMs to compete with the binding of the EIM, which explains the increased $^{15}$N R$_{1ρ}$ rates around the DLF/FxDxF motifs towards the C-terminal end of AP180$_{399-598}$ in the presence of AP2β2 mutants as compared to the presence of AP2β2 wild type. We then complemented our NMR experiments with ITC performed on AP180$_{399-598}$ into which the different AP2β2 mutants were titrated. Indeed, compared to the ITC experiments performed with the AP2β2 wild type, all mutants weaken binding to AP180 only mildly (Supplementary Fig. 14), in agreement with NMR spin relaxation performed on $^{15}$N labeled AP180$_{399-598}$ in presence of the different AP2β2 mutants. Interestingly, AP2β2 K842E and Y888A, as well as the AP2β2 wild type, interact with AP180$_{399-598}$ in an exothermic way, while AP2β2 Y815A interacts with AP180$_{399-598}$ in an endothermic fashion. Y815A is the only mutation in the sandwich domain, while both other mutations (K842E and Y888A) reside on the platform domain, suggesting that enthalpic and entropic contributions towards binding are distinct for the platform and sandwich domains of AP2β2.

In order to pinpoint the interaction site of the EIM onto the surface of AP2β2, we assigned $^{15}$N-$^{13}$C labeled and deuterated AP2β2 (Supplementary Figs. 15, 16), and titrated it with AP180$_{430-500}$, comprising only the EIM (thus excluding the C-terminal SLiMs, Fig. 7). Significant interaction is obvious from many peaks disappearing already at very low admixtures of AP180$_{430-500}$, in agreement with the reverse experiment, when AP180 was labeled (Fig. 4). We mapped chemical shift perturbations (CSPs) and intensity ratios throughout the titration and identified 5 distinct clusters of interactions along the sequence of AP2β2. When CSPs were mapped onto the AP2β2 crystal structure, a contribution of binding from the platform and the sandwich domain is clear, with the platform domain showing stronger peak intensity drops as compared to the sandwich domain (Fig. 7). AP180$_{430-500}$ comprises one single lysine residue close to its C-terminus, in the DLF motif that is part of the EIM. We thus used this property to position the C-terminus of AP180$_{430-500}$ with respect to the interaction surface on AP2β2 using cross-linking mass spectrometry. For this, we mixed AP180$_{430-500}$ and AP2β2 at equimolar concentrations and cross-linked the protein mixture using the lysine active cross-linker disuccinimidyl suberate (DSS). In the absence of trypsin cleavage sites within AP180$_{430-500}$, we cleaved the proteins using the endoproteinase Glu-C, which cleaves C-terminal of either aspartic or glutamic acid. To identify inter-molecular cross-links, the digested peptides were analyzed using tandem mass spectrometry. Out of, in total, 15 lysines contained in AP2β2, AP180$_{430-500}$ cross-linked specifically to two lysines in the platform domain that are in close spatial vicinity to each other (K854, and K935, Fig. 7C, D, Supplementary Table 4), arguing for a well defined orientation between AP180$_{430-500}$ and AP2β2 upon interaction, likely with residues 470–500 of AP180 contacting the platform domain and 435–445 contacting the sandwich domain of AP2β2.

## The concerted interaction of AP2β2:AP180$_{430-500}$ is highly dynamic

We then sought to determine whether both parts of the EIM (residues 435–455 and 470–500) were part of the same interaction process, or

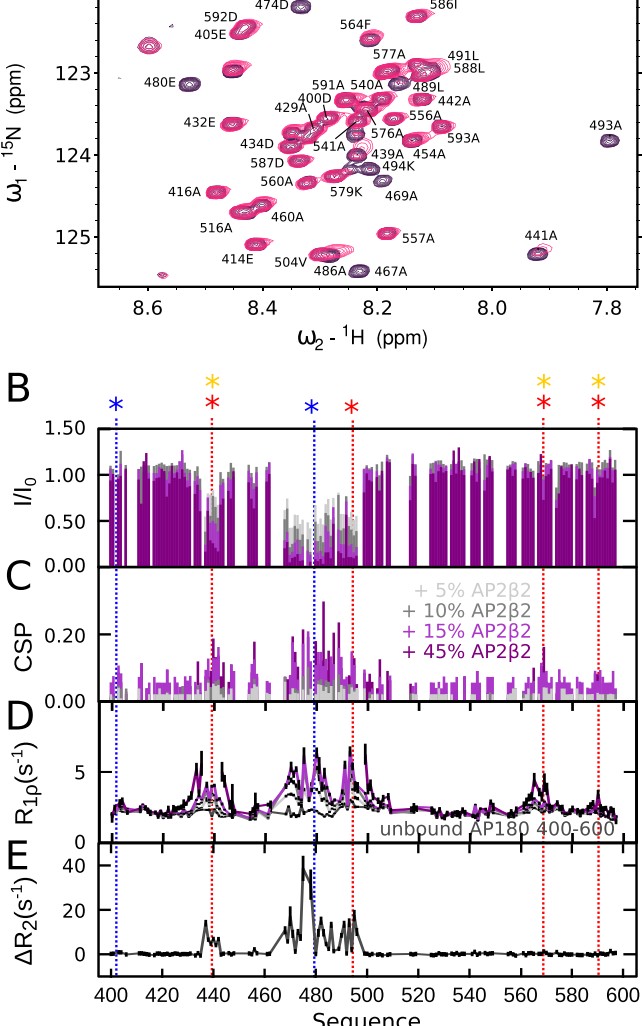

**Fig. 4 | Interaction of AP180 with APβ2. A** Spectrum of AP180$_{399-598}$ in the absence (purple) and the presence (pink) of 45% AP2β2. **B** Intensity ratios (I/I$_0$) of AP180$_{399-598}$ in the presence versus the absence of increasing concentrations of APβ2. **C** Combined chemical shift perturbations (CSPs) between the $^1$H-$^{15}$N HSQC spectra of AP180$_{399-598}$ alone and in the presence of increasing amounts of APβ2. **D** $^{15}$N R$_{1ρ}$ spin relaxation of AP180$_{399-598}$ in the absence and presence of increasing concentrations of AP2β2. The color code for **B**−**E** is displayed in **C**. **E** ΔR$_2$ as derived from a CPMG relaxation dispersion experiment from AP180$_{399-598}$ with 10% APβ2. Plotted is the difference between R$_2$ at CPMG frequencies of 31.35 Hz and 1000 Hz. All experiments were performed at a concentration of 100 μM AP180$_{399-598}$ and at a $^1$H Larmor frequency of 600 MHz. Errors are the sum of the errors from the respective R$_2$ values. Source data are provided as a Source Data file.

whether they interacted independently from each other. We thus tested whether a construct of AP180 comprising only residues 470−500 could compete off both parts of the interaction site. For this, we first added AP2β2 at a ratio of 10% to $^{15}$N labeled AP180$_{399-598}$. In comparison to AP180$_{399-598}$ alone, many peaks in the $^1$H-$^{15}$N HSQC spectrum had nearly disappeared. We then added unlabeled AP180$_{470-500}$ and assessed whether those peaks would reappear, which was indeed the case, albeit at a quite large excess of AP180$_{470-500}$ (Fig. 8A, Supplementary Fig. 17). Closer inspection as well as a plot of the intensity ratios and $^{15}$N R$_{1ρ}$ rates against the amino acids sequence reveals that upon addition of AP180$_{470-500}$, both parts of the EIM are displaced from interacting with AP2β2.

To asses a concerted interaction between parts 435−455 and 470−500 of the EIM further, and based on the rapid disappearances of

peaks from the $^1$H-$^{15}$N HSQC spectrum of AP180 in the presence of only small amounts of AP2β2, we speculated that AP180 might be in exchange between its unbound and its bound form on the micro- to millisecond timescale. We thus measured CPMG relaxation dispersion of AP180$_{399-598}$ with 10% of AP2β2. This NMR experiment uses a pulse train of different frequencies, which will refocus the exchange contribution towards the NMR signal (or the effective transverse relaxation R$_2^{eff}$) depending on the time scale of exchange compared to the frequency of the CPMG pulse train (ν$_{CPMG}$). In the presence of exchange on the micro- to millisecond timescale, the measured R$_2^{eff}$ for a given residue will depend on the refocusing frequency and allows, through fitting with a 1:1 binding model, to extract the percentage of bound population as well as the exchange rate (k$_{ex}$ = k$_{on}$ [AP2β2] + k$_{off}$). Our experiments indeed revealed a contribution of exchange in the micro- to millisecond time scale for residues across both parts of the EIM. CPMG curves for all 22 residues showing significant dispersion (i.e. dependence of R$_2^{eff}$ on ν$_{CPMG}$) across both parts of the EIM were fit together with a 2-state binding model and yielded an exchange rate of $662 \pm 35$ s$^{-1}$ and a percentage of bound population of $8.9 \pm 0.97$% (Fig. 8B, Supplementary Fig. 18). This result validated that both parts of the EIM interact with AP2β2 in a concerted way and further allowed us to calculate an equilibrium dissociation constant for the interaction between the EIM and AP2β2. The derived K$_D$ of 10 μM is in very good agreement with equilibrium dissociation constants obtained from ITC experiments with the different AP180 constructs (Fig. 5), validating again that it is indeed the EIM that determines the overall interaction between AP180 and AP2β2.

## Discussion

Intrinsically disordered proteins and regions take up vital functions in many different biological processes, despite (or because of) their lack of stable three-dimensional structure. In contrast to well folded proteins, IDPs exploit their highly flexible and disordered nature to function as entropic bristles, hub proteins bringing different partners together, initiators for liquid-liquid phase separation, recipients of post-translational modifications, and many more[30–32]. Their binding modes are often weak and transient, and short linear motifs, encompassing just a few amino acids (usually up to 15) are meanwhile well known to be responsible for many interactions between IDPs/IDRs and their folded partner proteins[27,33,34]. Remarkably, SLiMs do not always act alone, but often appear in clusters and engage in multivalent interactions[35,36], as for example in nucleo-cytoplasmic trafficking[25], cellular signaling[37], and membrane trafficking[38], including also clathrin mediated endocytosis[5,7,14,21,24].

In the early phases of clathrin mediated endocytosis, a complex network of IDR-mediated interactions relying on the presence of multiple SLiMs, also suspected to contribute to liquid-liquid phase separation[39–41], orchestrates the onset of vesicle uptake. Much information on binding and molecular coordinates of the bound complexes has been obtained from some of the SLiMs contained in endocytic IDRs[5,6,17,18,28] and has led to the development of consensus sequences and the perception that binding interactions can be predicted from the primary sequence[7,21]. Nonetheless, molecular resolution information has so far only been obtained from protein complexes comprising small peptides of only few residues in length derived from endocytic IDRs, with only a few exceptions, where larger IDR constructs have yielded precise information on binding, as for example binding of the clathrin associated sorting protein (CLASP) FCHO1 to the major adapter protein AP2[29], the binding of Stonin2 to the EH domains of Eps15[42], or the binding of a 60 residue long construct of AP180 to the clathrin heavy chain terminal domain[24]. Excising SLiMs from their native background can be a fruitful strategy to obtain structural information on binding, but neglects the context the motif is embedded in and the potential interplay between various SLiMs of the same or different kind[30,43].

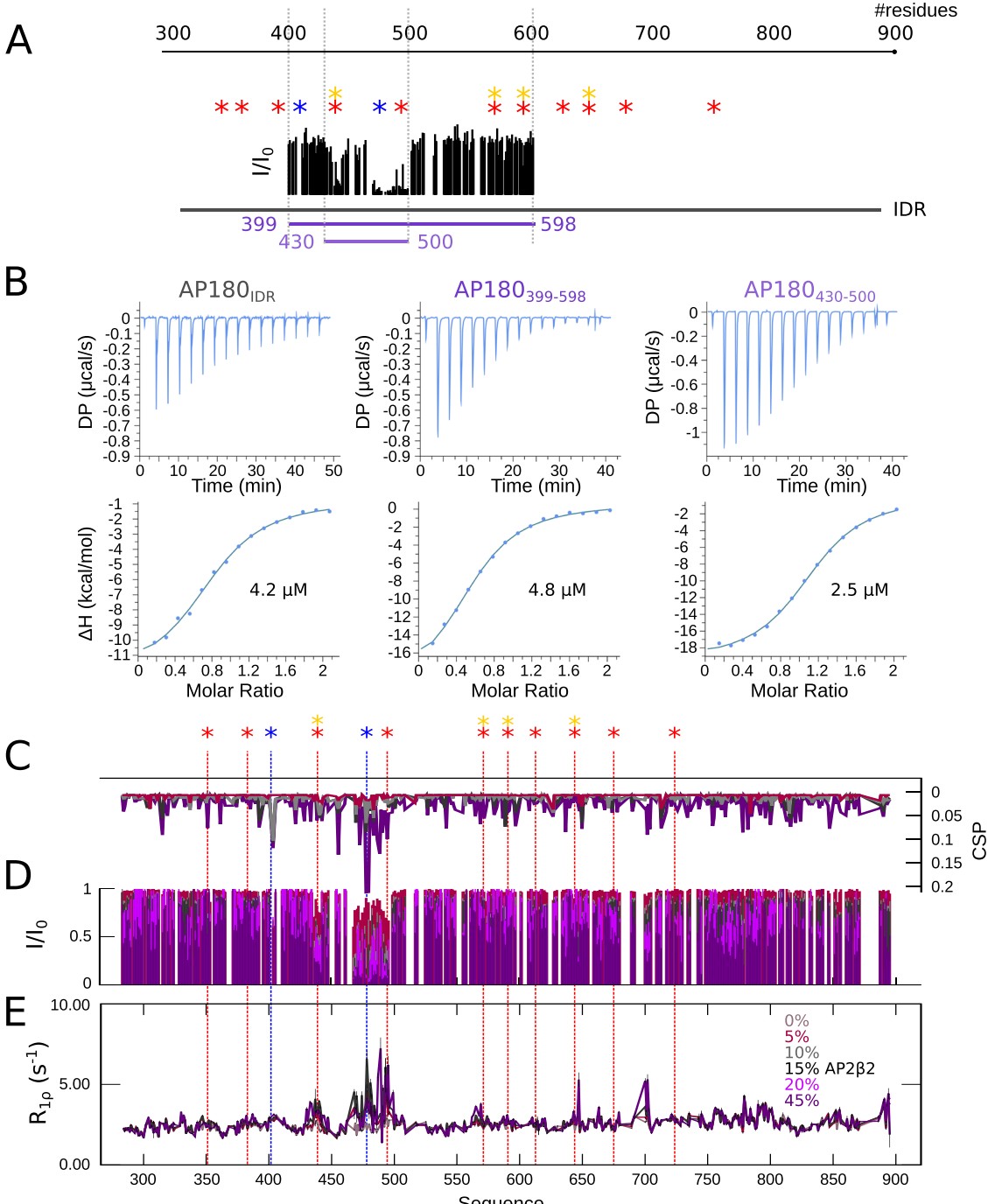

**Fig. 5 | Binding affinities between AP180 and APβ2. A** Scheme of AP180 constructs used for ITC analysis. Putative binding motifs are indicated as colored stars (color code as in Fig. 1). $I/I_0$ of AP180_{399-598} in the presence of 45% APβ2 versus the absence of APβ2 are indicated in black to illustrate the choice of the AP180 constructs. **B** ITC of AP180_{IDR}, AP180_{399-598} and AP180_{430-500} upon interaction with APβ2. Differential heating powers (DP) per injection are shown on top, enthalpy versus molar ratio of the interaction partners are shown in the bottom row. The data are fitted with a 1:1 binding model resulting in the affinities indicated in the respective graphs ($K_D$ values of single experiments are shown). **C** Absolute chemical shift perturbations (CSPs) of AP180_{IDR} in the absence versus the presence of increasing concentrations of AP2β2. **D** Intensity ratios of AP180_{IDR} in the presence of increasing concentrations of AP2β2 (I) versus its absence ($I_0$). **E** $^{15}$N $R_{1\rho}$ relaxation rates of AP180_{IDR} in the absence and presence of increasing concentrations of AP2β2. The color legend for **C**–**E** is displayed in **E**. The concentration of AP180_{IDR} was 100 μM throughout the NMR experiments. Errors were derived from the experimental uncertainty. Source data are provided as a Source Data file.

Using NMR spectroscopy, we investigated the conformational dynamics and interaction landscape of the full length IDR of AP180, an extremely long CLASP expressed in neurons. AP180 harbors an extended IDR of about 600 residues, and comprises a total of 10 putative clathrin binding motifs, distributed across 400 residues. In addition, multiple small AP2 binding motifs expected to bind to the α- and β2-appendage domains are dispersed across the sequence, and none of the predicted motifs is longer than 5 residues (FxDxF) in length. Our strategy using a divide and conquer approach and studying both different overlapping AP180 segments as well as the full AP180_{IDR} by NMR spectroscopy, allowed us to investigate its conformational sampling and binding to its diverse interaction partners on a residue by residue level.

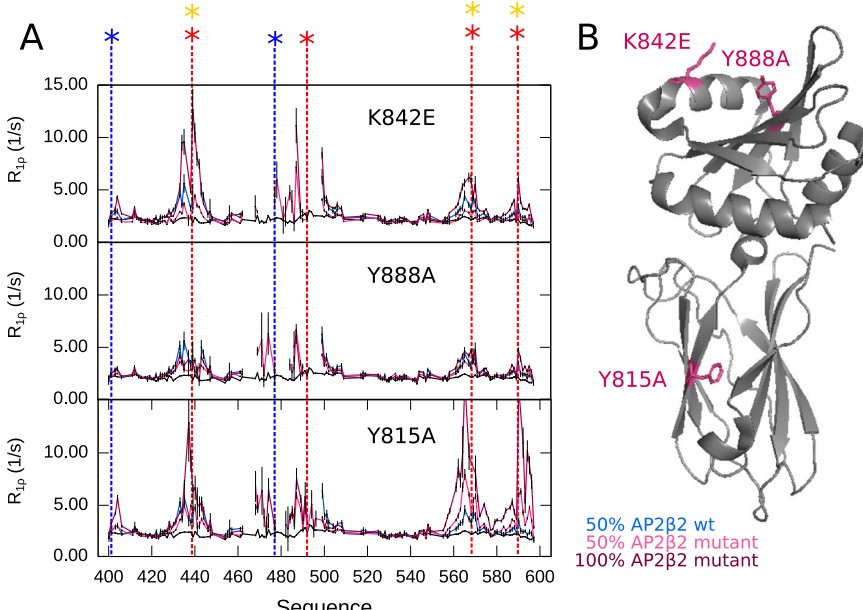

**Fig. 6 | Interaction of AP180₃₉₉₋₅₉₈ with AP2β2 mutants. A** $^{15}$N $R_{1\rho}$ relaxation rates of AP180$_{399-598}$ alone (black), with 50% AP2β2 wild type (blue) and 50% and 100% AP2β2 mutants (light and dark pink as indicated in the figure). Mutants are K842E, Y888A,Y815A (from top to bottom). Putative interaction motifs are indicated with colored stars, colors as in Fig. 1. AP180$_{399-598}$ was at a concentration of 100 μM and relaxation was measured at a $^{1}$H Larmor frequency of 600 MHz for all experiments. Errors were derived from the experimental uncertainty. **B** Structure of AP2β2 (PDB 1E42). Mutation sites are indicated as pink sticks. Source data are provided as a Source Data file.

AP180 appears to be unstructured across the entire length of its IDR, with very little signs of transient secondary structure, evident only around residues 380–400 and 750–780, which sample α-helical structures to a small extent. While a previous study has identified two DLF motifs to populate small β-turn like structures in the absence and presence of the clathrin heavy chain terminal domain using NMR NOESY experiments[24], DLL/DLF motifs do not seem to generally be present in a certain conformational environment, such as with α-helical or extended conformations, as suggested from secondary chemical shifts (Supplementary Figs. 7, 8). Protein backbone dynamics are reminiscent of those of a disordered chain, with only a mild increase towards the C-terminus of AP180.

Binding of the clathrin heavy chain terminal domain (CHC$_{TD}$) occurred around all DLL/DLF motifs within AP180$_{IDR}$ with little pre-ference for a certain motif (Fig. 2, Supplementary Fig. 9), in agreement with the previous observation that the number of DLL/DLF motifs present in N-terminally truncated AP180 IDR constructs affected overall clathrin binding in a linear way[14]. Residue-wise equilibrium dissociation constants calculated from AP180 $^{15}$N transverse relaxation yielded values in agreement with those previously determined using chemical shift titrations[24] in the hundreds of micromolar range. Bind-ing of the individual motifs seems to be independent from each other, judging from a decrease of the $^{15}$N $R_{1\rho}$ rates down to the values of the unbound AP180 between the interaction motifs, and in agreement with the observation that the affinities of one interaction motif within a two-motif construct does not change when the respective other motif is mutated[24]. The C-terminal region of AP180, which has previously been suggested to harbor a special interaction site with clathrin, investi-gated using peptide assays[44], shows mildly increased $R_{1\rho}$ relaxation rates upon interaction with CHC$_{TD}$ in the full AP180$_{IDR}$ but not the C-terminal segment AP180$_{720-898}$. This increase in $R_{1\rho}$ within AP180$_{IDR}$ could potentially be explained by slowed tumbling times of this mildly structured region in the presence of the full IDR when bound to CHC$_{TD}$. A direct interaction can, however, not be excluded.

Our data, measured on AP180 segments covering the full IDR, as well as on the AP180$_{IDR}$ itself, demonstrate that binding of CHC$_{TD}$ to AP180 is promiscuous and involves not only DLL/DLF motifs, but also DPF motifs, normally known to bind AP2α, or hydrophobic clusters (e.g. around residue 700, Supplementary Fig. 9). Promiscuous inter-action is particularly evident for AP2α binding, which shows an overall similar signature on AP180 as CHC$_{TD}$ binding, despite the fact that most motifs have in the past been classified as clathrin-, and not AP2-binding. Affinities between AP2α and AP180 are overall comparable to those between CHC$_{TD}$ and AP180. As for CHC$_{TD}$ binding, interaction is dispersed across the entire AP180$_{IDR}$ and shows no sign of a region that would be particularly important for AP2α binding.

The scenario is different for binding of AP2β2, where we identified a distinct and potent binding region within AP180. This region is of extended nature, spreading over in total 70 residues and showing intermediate exchange characteristics in the form of CPMG relaxation dispersion across as many as 22 non-overlapped residues (Figs. 4–8, Supplementary Figs. 12–14, 17, 18). To set this interaction site apart from the usually observed SLiMs, we termed it an Extended Interaction Motif (EIM). The EIM is split in two parts, residues 435–455 and 470–500 that both show signs of intermediate exchange and whose dispersion curves could be fitted together in a global fit, yielding an exchange rate between the bound and the unbound state of AP180 of 662 s$^{-1}$. This result as well as a competition assay that showed dis-placement of the whole EIM by only the region encompassing AP180 470–500, demonstrates a concerted interaction of the two parts of the EIM. The affinity for a 1:1 binding reaction was determined to be 10 μM using the fits of the CPMG relaxation dispersion experiments, which shows very good agreement with ITC data (5.1 ± 0.42 μM). The same affinity is maintained within error for the interaction between AP180$_{IDR}$ and AP2β2, which testifies that the EIM determines the overall inter-action between the two proteins – an observation that can equally be drawn from spin relaxation experiments of AP180$_{IDR}$ in the presence of AP2β2 (Supplementary Fig. 12).

The presence of the EIM is remarkable for several reasons. Firstly, it shows an affinity towards AP2β2 that is not normally observed between SLiM based interactions with their folded partner proteins, and only few endocytic CLASPs show higher affinities than those

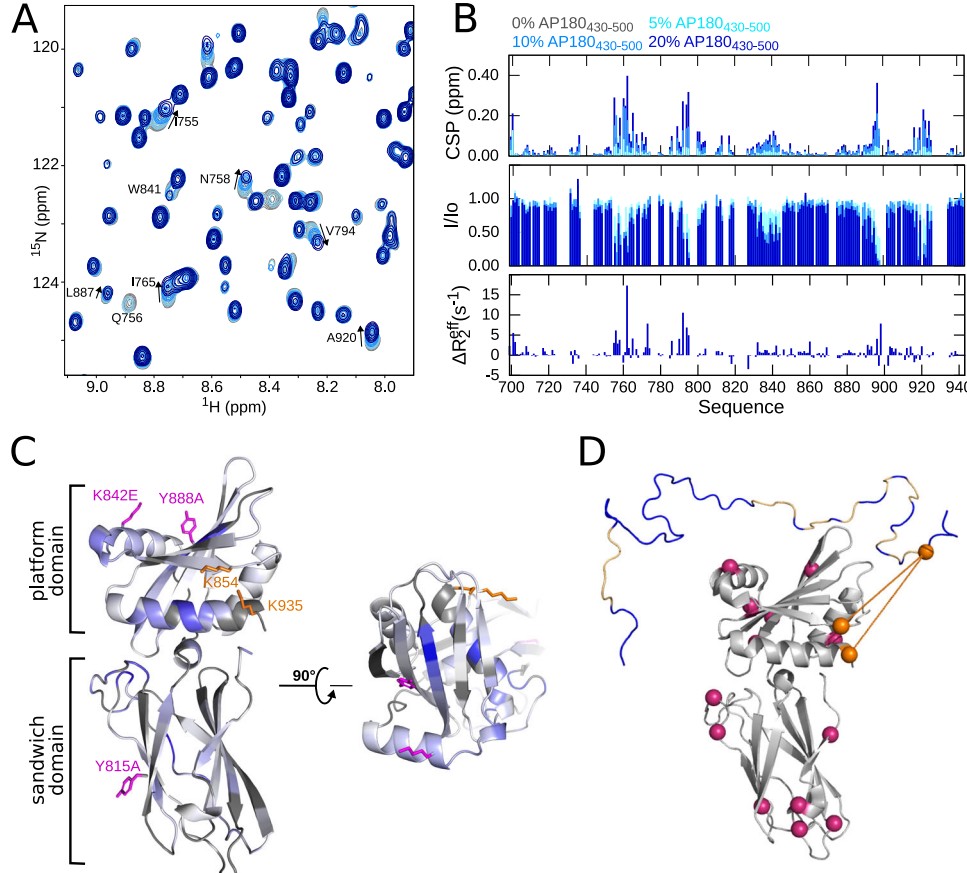

**Fig. 7 | Interaction of AP2β2 with AP180$_{430-500}$. A** Zoom into $^1$H-$^{15}$N Transverse relaxation-optimized spectroscopy (TROSY) spectra of $^{15}$N, $^2$H labeled AP2β2 (200 μM) in the absence and the presence of 5%, 10%, and 20% AP180$_{430-500}$ (color legend as in **B**). The assignment is annotated for residues that experience a change upon interaction. **B** Chemical shift perturbations (CSP) and intensity ratios (I/I$_0$) in the presence of different concentrations of AP180$_{430-500}$ (color legend on top). $\Delta R_2^{eff}$ is derived from a CPMG relaxation dispersion experiment from AP2β2 in the presence of 20% AP180$_{430-500}$ and describes the difference between $R_2$ at CPMG frequencies of 31.35 Hz and 1000 Hz. **C** Structure of AP2β2 (PDB 1E42). Chemical shift differences are annotated in different shades of blue at 20% AP180$_{430-500}$ concentration. Gray represents unassigned residues and prolines. Literature-known binding mutants are annotated in pink. Lysines that cross-link with AP180$_{430-500}$ are annotated in orange. **D** Cross-linking network between AP180$_{430-500}$ and AP2β2. The PDB structure of AP2β2 (1E42) is shown in gray and one conformation of AP180$_{430-500}$ is shown in blue. All lysines in both proteins are illustrated as balls. Those not involved in inter-molecular cross-links are colored pink. Lysines involved in inter-molecular cross-links between AP180$_{430-500}$ and AP2β2 are shown in orange. Cross-links are illustrated as orange lines. Residues showing dispersion in a CPMG relaxation dispersion experiment (see Fig. 4E) are colored in light orange on AP180 and illustrate the residues in contact with AP2β2. Note that the organization of the two proteins in the figure does not reflect an actual binding mode, and only serves the purpose of illustrating inter-molecular cross-links detected my mass spectrometry. Source data are provided as a Source Data file.

observed between AP180 and AP2β2. Among those count the interaction between Stonin2 and the EH2 domain of the CLASP Eps15[42], as well as the interaction between the Eps15 IDR and the AP2 α-appendage domain[5], which both have sub-micromolar affinities. For both protein complexes, two sites on the folded protein (EH2 and AP2α, respectively) together with two known SLiMs on the IDRs have been shown to participate in the interaction. We also observe the binding of several interaction sites (or a larger interaction surface) on the folded partner protein, explaining the significantly higher affinity. We were indeed able to map the extended interaction surface on both AP180 and AP2β2 using NMR chemical shift mapping together with $^{15}$N spin relaxation and relaxation dispersion, clearly delineating what residues are involved in the interaction from both sites. Cross-linking mass-spectrometry exploiting the fact that AP180$_{430-500}$ has only one single lysine at its C-terminus, right at the edge of the EIM, allowed us to propose an orientation of AP180$_{430-500}$ on the surface of AP2β2, whereby the C-terminal end of the EIM would be in close contact with two lysines (K854 and K935) in the AP2β2 platform domain. Particularly, we have now been able to observe this comparatively strong interaction from the side of AP180's (full) IDR, demonstrating that, even through the EIM encompasses three SLiMs in total, many other residues are involved in the binding, which indeed prompted us to consider the interaction as an extended interaction rather than an interaction using multiple SLiMs. This notion is strengthened by the fact that all residues within the EIM act in a concerted way and that other clusters of SLiMs together do not achieve comparable affinities. In this context, it is particularly noteworthy that all DLL/DLF motifs together do not lead to a measurable affinity between AP180 and CHC$_{TD}$ in an ITC experiment.

While many interactions between CLASPs and AP2 have been characterized using the α-appendage domain, it is known that the β2-appendage domain has slightly different binding preferences. A few CLASP peptides, such as β-arrestin and ARH, have shown to bind to AP2β2 with roughly similar affinities than AP180[6], apparently in the absence of an EIM. Even though crystal structures of these complexes are available[6], an IDR-centered view on these interactions would be of high interest in view of the EIM discovered in AP180. Notably, no obvious sequence similarity can be observed between the devised β-arrestin peptides and the EIM contained in AP180.

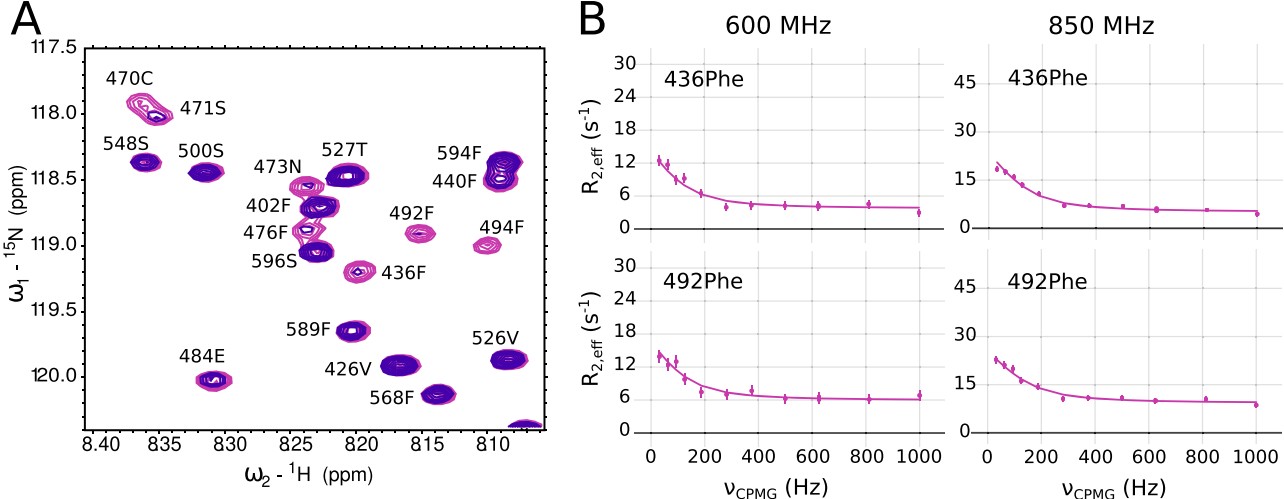

**Fig. 8 | Cooperative interaction of the entire AP180 binding site with AP2β2.**
**A** Zoom into a $^1$H-$^{15}$N HSQC spectrum of AP180$_{399-598}$ in the presence of 10% AP2β2 (purple) and with the addition of 450% AP180$_{470-500}$ (pink). The concentration of AP180$_{399-598}$ was 100 μM. **B** $^{15}$N CPMG relaxation dispersion of $^{15}$N AP180$_{399-598}$ with 10% AP2β2 at a concentration of 100 μM. Shown is one residue of the main interaction site (F492) and one residue from the side lobe (F436) at field strengths of 600 and 850 MHz $^1$H Larmor frequency. Errors were determined based on standard deviations of repeat measurements.

Importantly, beside the EIM, AP180 contains numerous SLiMs that bind to both AP2 appendage domains, as well as the clathrin heavy chain terminal domain with affinities orders of magnitudes lower. Nevertheless, we see that these SLiMs do engage in binding, which becomes particularly obvious when EIM binding is weakened through mutations in AP2β2 (Fig. 6) or by comparison of constructs in AP180 that do or do not contain the EIM (Supplementary Fig. 12, compare AP180$_{540-740}$ and AP180$_{IDR}$). In addition, once proteins are concentrated at the endocytic pit and prepared for vesicle uptake, local protein concentrations may rise and the more weakly binding SLiMs gain importance as the EIM is more and more occupied. Our data thus advocate for a model, whereby both the higher as well as the lower affinity interactions are important at different stages of endocytosis, as illustrated in Fig. 9: Rather than requiring a clear order of binding events, such as thought of in a signaling cascade, local concentrations of AP180 and its partners will determine the likelihood of one or the other interaction to take place: A single binding event is always of stochastic nature and how likely it is will be determined by its binding affinity. At a low concentration of endocytic partners (e.g. AP2), for example very early during the endocytic event, the strong interactions will dominate the process and contribute to keeping/bringing the machinery together. At higher concentrations of partners, after many of the endocytic components have been recruited, the weak SLiM dominated interactions will start playing a bigger role and are expected to allow for sufficient freedom for the different players to rearrange in space until vesicle uptake can take place. As in other multimotif systems, these weak interactions enable rapid and efficient movement, which is likely crucial for a timely uptake of endocytic vesicles.

Finally, it is worth noting that the EIM-interacting region resides in the AP2 β2-appendage domain, which contains a clathrin box motif in its linker towards the core domain. It is thus appealing to speculate that the EIM and clathrin-box together with CHC$_{TD}$ binding to SLiMs are the molecular determinants that enable effective clathrin assembly, and additional molecular studies on these interactions will certainly further its understanding.

## Methods
### Protein constructs
H6-AP180 was a gift from Ernst Ungewickell (Addgene plasmid # 63006; http://n2t.net/addgene:63006; RRID:Addgene_63006)[45]. The different AP180 constructs used in this study were cloned from H6-AP180 into either a modified pET-28a vector with an N-terminal GB1 solubility tag pET-28a-GB1-6His (470–500, 430–500, 281–500, 721–898, 281–898, 399–598/IDR) or pET-41c(+) (540–740, 471–701) vector, leading to the expression of a TEV (tobacco etch virus) cleavable GB1-6His-TEV site-AP180-6His (281–500, 720–898, 281–898/IDR, flanking residues are GHM at the N-terminus and LEHHHHHH at the C-terminus), GB1-6His-TEV site-AP180 (470–500, 430–500, 399–598, flanking residues GHM are present at the N-terminus) or AP180-6His constructs (540–740, 471–700, flanking residues are M at the N-terminus and LEHHHHHH at the C-terminus). The gene of AP180 used in this study corresponds to the rat isoform X3, which carries the RefSeq protein ID XP_006243556.1. Ala538 is deleted in our construct compared to the RefSeq reference.

GST-beta2(616–951) was a gift from Stephen Royle (Addgene plasmid # 100743; http://n2t.net/addgene:100743; RRID:Addgene_100743)[46]. The protein constructs 617–951 and 714–951 of the bovine protein (Uniprot ID Q08DS7) were cloned into the pET-28a-GB1-6His vector, leading to TEV cleavable constructs GB1-6His-TEV site-AP2β2-6His. Throughout the interaction experiments AP2β2$_{617-951}$ was used if not otherwise noted. Experiments with NMR labeled AP2β2, as well as cross-linking mass spectrometry experiments, were performed on AP2β2$_{714-951}$, devoid of the linker otherwise connecting the AP2β2 core and appendage domain. We validated on key experiments that interactions with AP180 were independent of whether the linker was present or not. Point mutations to yield AP2β2 K842E, Y888E, Y815E were inserted using site-directed mutagenesis. Note that, even though the bovine protein contains a small insert in the AP2β2 linker as compared to the human form, changing the effective residue numbers, all numbers throughout the main text refer to the human form for comparability with the literature.

Genes of mouse AP2α (residues 701–938 of Uniprot ID P17427.2) and bovine CHC$_{TD}$ (residues 1–363 of Uniprot ID P49951.1) cloned into pET-28a(+) and yielding constructs AP2α/CHC$_{TD}$-TEV-6His were purchased from Twist Bioscience.

### Protein expression
The *E. coli* strain Rosetta™ (DE3) (Sigma-Aldrich) or BL21-AI™ (Invitrogen, for AP180$_{IDR}$) were used for protein expression. Cultures were grown at 37 °C and in lysogeny broth (LB) medium until an optical density (OD) at 600 nm of around 0.6–1, after which expression was

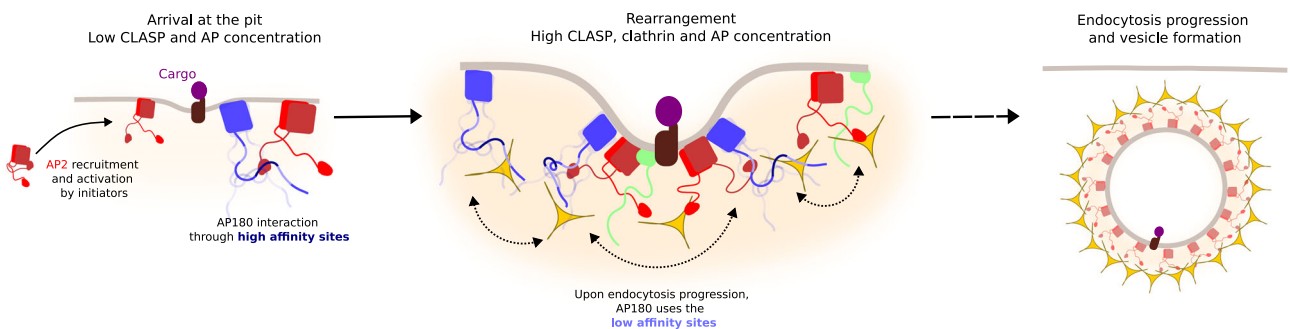

**Fig. 9 | High and low affinity interactions together determine progression of endocytosis.** At the beginning of the endocytic event, protein concentrations at forming pits are small (left) and favor thus high affinity interaction, such as the one included in AP180. At higher concentrations of AP180 and its partners (clathrin and AP2), all high affinity sites are occupied and low affinity sites become important. They are of transient nature and allow rearrangement (middle) to finally enable an arrangement that can progress towards a clathrin coated vesicle (right).

induced with 1 mM isopropyl-β-D-thiogalactopyranoside (IPTG). Expression was allowed to continue at 20 °C over night, after which cultures were harvested. Proteins labeled for NMR ($^{15}$N and/or $^{13}$C) were expressed in M9 minimal medium supplemented with 2 g/L $^{13}$C-glucose (Eurisotop) and/or 1 g/L $^{15}$N-NH$_4$Cl (Eurisotop). For the production of deuterated AP2β2, M9 medium was prepared in D2O (Eurisotop).

### Protein purification
Cells were lysed by sonication in lysis buffer (20 mM Tris, 150 mM NaCl pH8 supplemented with a Roche Ethylenediaminetetraacetic Acid (EDTA)-free protease inhibitor cocktail (Sigma-Aldrich)). Purification was achieved in a 2-step process: through standard Nickel purification, followed by a Size-Exclusion Chromatography (SEC). The cleared lysate was loaded onto a self-packed nickel column (Fisher Scientific), after which the column was washed in lysis buffer supplemented with 20 mM imidazole. The protein was eluted in lysis buffer comprising additional 400 mM imidazole. The elution fraction containing the pure protein (validated by SDS-PAGE and Coomassie staining) was dialyzed over night at 4 °C in 500 mL lysis buffer with 5 mM β-mercaptoethanol and 1 mg TEV protease. Further purification through SEC was performed on a Superdex$^{TM}$75 or 200 column (Cytiva) equilibrated in NMR buffer (50 mM Na-phosphate pH 6, 150 mM NaCl, and 2 mM dithiothreitol (DTT)). Fractions containing pure protein were concentrated and frozen. Final protein concentrations were determined by absorbance at 280 nm. For constructs lacking aromatic residues, concentration was determined by the Bicinchoninic Acid method using the Thermo Pierce BCA Assay Kit (Thermo Fisher Scientific, Waltham, MA).

A second nickel purification was performed for AP180$_{430-500}$ and AP180$_{470-500}$, which do not comprise a C-terminal His-Tag, after TEV cleavage and over night dialysis. The purified protein was collected in the flow through, concentrated and further purified by SEC.

In contrast to all other AP180 segments, the lysis, wash and elution buffers of AP180$_{IDR}$ contained additional 4 M urea.

SEC on CHC$_{TD}$ was performed using 50 mM Na-phosphate pH 7.5, 150 mM NaCl, and 2 mM dithiothreitol (DTT).

Protein purity was validated using SDS-PAGE and NMR spectroscopy (Supplementary Figs. 2–6 and 15).

### NMR spectroscopy
NMR experiments were acquired on Bruker spectrometers at the Institute of Structural Biology (IBS), Grenoble, France ($^1$H frequencies of 600, 700, 850 MHz), and at the Leibniz-Forschungsinstitut für Molekulare Pharmakologie (FMP), Berlin, Germany ($^1$H frequencies of 600, 750, 900 MHz) using the acquisition software TopSpin 3.5. The spectrometers were equipped with either room-temperature- (600, 750 MHz) or cryo-probes (600, 700, 850, 900 MHz). All experiments

were measured in NMR buffer (50 mM Na-phosphate pH 6, 150 mM NaCl, and 2 mM DTT) at 25 °C if not otherwise noted.

### Backbone $^{15}$N and $^{13}$C assignments
The spectral assignments of the different AP180 $^{13}$C- and $^{15}$N-labeled constructs were obtained at a $^1$H frequency of 600 MHz using using band-selective excitation short transient–transverse relaxation-optimized spectroscopy (BEST-TROSY) triple resonance experiments correlating Cα, Cβ, and CO resonances (BT-HNCO, BT-HNcaCO, BT-HNcoCA, BT-iHNCA, BT-HNcocaCB, BT-HNcaCB)[47]. Processing of the spectra was done with NMRPipe[48], automatic assignment was performed with the program MARS[49] and manually curated. Spectral analysis was performed using Sparky[50]. Secondary structure propensities[23] and secondary chemical shifts were derived using random coil values from refDB[22].

Backbone assignment of AP2β2$_{714-951}$ was performed with triple labeled ($^{15}$N, $^{13}$C, $^2$H) protein at 25 °C and 32 °C at a protein concentration of 420 μM and using non-uniform sampling (NUS). The spectra were processed with qMDD[51] and NMRPipe[48] and assigned using a combination of manual assignment in CcpNmrAnalysis v 3.1.1[52] and automated assignment with MARS[49].

### $^{15}$N relaxation, relaxation dispersion, and titrations
Utilizing the procedures detailed in the main text and figures, we investigated the interactions involving AP180 and its associated partners. Extraction of peak intensities ($I$) and $^1$H, as well as $^{15}$N chemical shifts, was carried out from $^1$H-$^{15}$N HSQC or $^1$H-$^{15}$N TROSY (AP2β2) spectra. Combined chemical shift perturbations (CSPs) were calculated using

$$CSP = \sqrt{\left(\delta^1H \cdot 6.5\right)^2 + \left(\delta^{15}N\right)^2} \tag{1}$$

Titration of the different AP180 segments with CHC$_{TD}$ was performed by titrating unlabeled CHC$_{TD}$ in 50 mM Na-phosphate pH 7.5, 150 mM NaCl, and 2 mM dithiothreitol (DTT) into $^{15}$N labeled AP180 in NMR buffer (pH 6). Control experiments were performed through titrating the same amount of pH 7.5 buffer into AP180 segments to exclude CSPs through altered buffer conditions.

All other proteins were in NMR buffer and no particular caution had to be taken in the titration experiments.

$^{15}$N R$_{1\rho}$ relaxation rates[53] were assessed at 600, 750 MHz and 900 MHz, maintaining a concentration of 100 μM of [U-15N]-labeled or [U-15N, U-13C]-labeled AP180 (IDR or segments) unless otherwise noted, and with varying amounts of partner, as outlined in the figures. The spin-lock field was set to 1500 Hz, and 7 delays, between 10 and 230 ms, were used to sample the decay of magnetization. Error bars

were estimated from Monte Carlo simulations of the experimental uncertainty.

For fast exchange behavior between the unbound and bound AP180, residue specific $K_D$ values were approximated from the measured $^{15}N$ $R_{1\rho}$ relaxation rates. At a spin-lock field of 1500 Hz and only small changes in $R_1$ throughout the titration, the following relation can be approximated under the condition of weak binding[25,54]:

$$R_{1\rho} = \frac{[B_{tot}]}{K_D + [A_{tot}]}\left(R_{1\rho}^B - R_{1\rho}^A\right) + R_{1\rho}^A. \tag{2}$$

We measured $R_{1\rho}{}^B$ of the respective binding partners. The residue-wise $K_D$ value can then be obtained from a linear fit of AP180 $^{15}N$ $R_{1\rho}$ rates plotted against the concentration of added interaction partner ($CHC_{TD}$ or AP2$\alpha$).

Relaxation dispersion experiments were conducted at 600 and 850 MHz, employing 14 CPMG frequencies ranging from 31 to 1000 Hz, with a constant-time relaxation of 32 ms. Errors were determined based on standard deviations of repeat measurements. The minimum error was set to 0.5. This was carried out on 100 μM $^{15}N$ AP180$_{399-598}$ with 10 μM unlabeled AP2$\beta$2. Data encompassing all 22 non-overlapped residues showing dispersion were collectively fitted using ChemEx (https://github.com/gbouvignies/chemex) using a two-state exchange model. The population of AP180 bound to AP2$\beta$2 ($p_B$) and the exchange rate ($k_{ex}$) were derived from the fitting. The $K_D$ value was calculated from the concentrations of both proteins used in the CPMG experiment and $p_B$ under the assumption of a 1:1 binding stoichiometry.

For cloning reasons, plots of NMR parameters along the sequence of AP180 are shifted by +1 compared to the sequence of the full length AP180.

### Isothermal titration calorimetry

Binding of AP180 to AP2$\beta$2 was investigated by Isothermal Titration Calorimetry on a iMicroCal iTC200 from Malvern at 25 °C. The experiments were conducted on 0.03 mM AP180$_{IDR}$, AP180$_{399-598}$, and AP180$_{430-500}$ in the microcalorimeter cell, adding 2 μl of AP2$\beta$2 (0.3 mM) at each titration step. The buffer was 150 mM NaCl, 50 mM Na-phosphate, and 0.2 mM TCEP (pH 7.5). Titration of 0.3 mM AP2$\beta$2 into buffer was recorded as a negative control and showed a flat curve. For binding of AP180$_{399-598}$ to AP2$\beta$2 mutants, 0.04 mM AP180$_{399-598}$ was kept in the microcalorimeter cell. The titration curves were fitted to the experimental data using the Origin version 7.0 software from MicroCal.

### Cross-linking mass spectrometry

Equal molar mixed AP180$_{430-500}$ and AP2$\beta$2$_{714-951}$ (100 μM each) in 50 Mm Na-phosphate buffer was cross-linked with disuccinimidyl suberate (DSS, Cayman Chemical) using 1 mM final concentration for 15 min in room temperature. The cross-linking reaction was quenched with 20 mM Tris-HCl (Carl Roth GmbH) for 15 min. For digestion, cross-linked proteins were denatured with 6 M Urea (Carl Roth), reduced with 10 mM dithiothreitol (DTT, VWR Life Science) at 37 °C for 45 min, and alkylated with 40 mM chloroacetamide (CAA, Sigma-Aldrich) at 37 °C for 45 min in the dark. After diluting to 2 M Urea with 50 mM triethylammonium bicarbonate buffer (TEAB, pH 8.0, Sigma-Aldrich), proteins were digested overnight with Glu-C (Promega) using an enzyme-to-protein ratio of 1:50 (w/w) at 37 °C. StageTips filled with $C_8$ material (CDS Analytical) for peptide desalting. Self-diluted 0.1% (v/v) Formic Acid (FA, VWR chemical) was used for the StageTips condition/peptides clean and 0.1%FA/80% Acetonitrile (v/v) (ACN, VWR chemicals) was used for StageTips activation/peptide elution. Sample was then dried using a SpeedVac SPD1030 (Thermo Fisher Scientific).

Peptides were resuspended in 0.1% (v/v) Formic Acid (FA, biosolve Chimie SARL) for LC-MS/MS analysis. The analysis was performed on

Vanquish Neo UHPLC system coupled with Orbitrap Exploris 480 (Thermo Fisher Scientific). Peptide separation was performed using a 50 cm reverse-phase column (in-house packed with Poroshell 120 EC-$C_{18}$, 2.7 μm) with 180 min gradient, 50 °C. The buffer A and B were using 0.1% FA (v/v) and 0.1% FA/80% ACN in LC system for the separation with 250 nL/min. The separation was started at 0.0 % buffer B, followed by linear increasing to 5% buffer B. The gradient was then slowly altered to 45% buffer B within 160 min and ended up with 99% buffer B washing for another 15 min. The system was protected by an installed pre-column (75 μm × 2 cm Acclaim PepMap 100 nanoViper filled with $C_{18}$, 3 μm, 0.1 nm, Themo Fisher Scientific). MS parameters in Data-dependent acquisition (DDA) mode were used as follows: 2 s Cycle time; 60 s Exclusion duration; 375–1400 m/z scan range; Normalized MS1 AGC 300%, 2200 V in positive mode for emitter; MS1 resolution 120,000; MS2 resolution 30,000; charge state 4–8 enabled for MS2; higher-energy collisional dissociation (HCD) 30%; FAIMS CV −40/−50/−60 V. MS acquisition was controlled by using Thermo Scientific Xcalibur 4.5.474.0.

Raw data were searched against a fasta database comprised of AP180 and AP2$\beta$2 sequences using pLink 2.3.11. Searching parameters were set as follows: 20 ppm for MS1 spectra, 20 ppm for MS2 spectra; 3 for the maximum miss sites; the peptide mass range was set from 600 to 6000; the peptide length was set from 6 to 60; Oxidation on Methionine was set as the variable modification. Searched result was separately filtered for intra-link and inter-link at 1% false discovery rate. One sample ($n = 1$) was analyzed.

### Reporting summary

Further information on research design is available in the Nature Portfolio Reporting Summary linked to this article.

## Data availability

Data supporting the findings of this paper are available from the corresponding author upon request and can be found in the Source Data file. Cross-link MS data generated in this study have been deposited on proteomeXchange under the accession code PXD049300. NMR assignments generated in this study have been deposited in the Biological Magnetic Resonance Bank under accession numbers 52320 (AP2$\beta$2), 52322 (AP180$_{281-500}$), 52323 (AP180$_{399-598}$), 52324 (AP180$_{471-700}$), 52325 (AP180$_{540-740}$), and 52326 (AP180$_{720-898}$).The PDB entry of AP2$\beta$2 used in this work is: 1E42 (Beta2-adaptin appendage domain, from clathrin adapter AP2). Source data are provided with this paper.

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

## Acknowledgements

We thank all members of the Milles group for fruitful discussions and critical proofreading. We thank A. Papagiannoula for recording AP2β2 assignment spectra, M.R. Jensen for providing the pET-28-GB1 plasmid, M. Blackledge and M.R. Jensen for providing analysis scripts, N. Trieloff and M. Beerbaum for technical assistance on the NMR spectrometers, and H. Nikolenko for assistance with ITC acquisition. This work used the platforms of the Grenoble Instruct-ERIC center (ISBG; UAR 3518 CNRS-CEA-UGA-EMBL) within the Grenoble Partnership for Structural Biology (PSB), supported by FRISBI (ANR-10-INBS-0005-02) and GRAL, financed within the University Grenoble Alpes graduate school (Ecoles Universitaires de Recherche) CBH-EUR-GS (ANR-17-EURE-0003), and with financial support from the TGIR-RMN-THC Fr3050 CNRS. We thank Caroline Mas for assistance and access to the biophysics platform. The Institut de Biologie Structurale acknowledges integration into the Interdisciplinary Research Institute of Grenoble. This work was supported by the Leibniz-Forschungsinstitut für Molekulare Pharmakologie (FMP) (to S.M., F.L. and P.S.). This project has received funding from the European Research Council (ERC) Starting Grant MultiMotif to S.M. under the European Union's Horizon 2020 research and innovation program (grant agreement no. 802209). We also acknowledge funding from the French National Research Agency (ANR) through an ANR T-ERC MultiMotif (ANR-17-ERC3-0004) and through the framework of the "Investissements d'Avenir" program (ANR-15-IDEX-02), DisRegulate (to S.M.).

## Author contributions

S.M. conceptualized the study; M.T., K.M., S.N.F., S.M. prepared samples; S.N.F., C.A.E.R., I.M.V., P.S., S.M. performed NMR experiments and analysis; P.L.J., F.L. performed MS experiments and analysis; M.T., C.A.E.R. and I.M.V. performed ITC experiments; S.M. and S.N.F. wrote the manuscript with help from all authors.

## Funding

## Competing interests

The authors declare no competing interests.
