## [Peer Review File · Nature Communications]

An extended interaction site determines binding between AP180 and AP2 in clathrin mediated endocytosisREVIEWER COMMENTS

Reviewer #1 (Remarks to the Author):

AP180 is a clathrin assembly protein with a very large intrinsically ordered domain (~60 kD) that plays a major role in assembling the complex network of endocytic proteins which underlies clathrin mediated endocytosis. The authors utilized NMR structural mapping to characterize interactions between AP180 and clathrin TD, as well as AP180 and AP2. Indeed they provide strong experimental support for the idea that the intrinsically disordered domain of AP180 contains a large number of clathrin binding sites centered around a DLL/DLF motif. In addition, they also find interactions between clathrin TD and DPF and FxDxF motifs on AP180, motifs previously thought to only be involved in AP2 binding. They go on to characterize interactions between AP180 and the alpha appendage domain of AP2. While indeed they find that AP2 alpha binds to DPF and FxDxF motifs on AP180 as expected, they also find promiscuous binding of AP2 alpha to the clathrin TD binding DLL/DLF motifs of AP180. While these are all weak interactions, something that is believed to be important for the necessary rearrangements to take place as clathrin coated vesicles are forming, they also identified a stronger interaction between AP180 and the beta appendage domain of AP2. While interactions between AP180 and either Clathrin TD or AP2 alpha are in the high micromolar to millimolar range, interactions between AP180 and AP2 beta are in the low micromolar range. This is novel, and informs the mechanism of protein network recruitment to nascent clathrin coated pits.

The strength of this paper lies in their exploitation of NMR spectroscopy to study these interactions in the context of the entire IDP domain of AP180. While some of what they found was expected from previously published work, it is very significant to see these ideas tested experimentally using an intact protein domain. For example, while no binding coupled folding was found in a previous NMR study of clathrin TD binding to a 57aa fragment of AP180 (citation 22 in this paper), some had questioned whether this result would hold up when the entire 60kD IDP domain of AP180 was utilized. Another strength of this paper is that it contributes to both the field of clathrin mediated endocytosis/vesicle trafficking, as well as to our understanding of intrinsically disordered protein domains.

The study is very well executed, and of extremely high technical proficiency. My only suggestion for improvements are editorial. It would be useful to the reader if the locations of all the binding sites identified are indicated more clearly in the tables. For example, in supplementary table 1, the residue numbers should be indicated at the start and end of each short sequence shown, to allow the reader to better integrate this study with previously published work.

Reviewer #2 (Remarks to the Author):

In this exciting manuscript from the Milles group, the authors present a tour-de-force characterization of the intrinsically disordered clathrin associated sorting protein AP180 and its interactions with various components of the endocytic machinery. Using solution NMR spectroscopy complemented with isothermal calorimetry and crosslinking mass spectrometry, the authors demonstrate that the long C-terminal IDR of AP180 is disordered in isolation and remains largely disordered and dynamic in complexes with clathrin and the α and $\beta 2$ appendages of the endocytic adaptor complex AP2. A particularly exciting finding of this study is that AP180 utilizes different interaction modes for different partner molecules. AP180 engages clathrin and the AP2 α appendage through a series of weak binding motifs that bind in a somewhat promiscuous manner, whereas a higher affinity interaction between AP180 and the AP2 $\beta 2$ appendage is mediated by a 70 residue 'extended interaction motif'. From these findings, the authors propose a model whereby local concentrations of endocytic proteins at various stages of the endocytic process will bias the AP180 complexes that form and contribute to progression of endocytosis. The results are well-presented, the conclusions are well-supported by the data, and the suggestions included below are minor and are not intended to interfere with publication of this important work.

1. Several sections of the manuscript could be clarified for a non-NMR audience.

- The authors reference the 'on-off binding rate' in the introduction. Later in the manuscript it becomes clear that this is the k_{ex} value determined from fitting the CPMG relaxation dispersion data. Some explanation of the relationship between k_{ex} and the on and off rate constants would be helpful.
- SSP values are presented in many figures and mentioned throughout the text, but they are not thoroughly explained. It would be useful to explain how these values can be interpreted. What do the values mean (sign and magnitude)?
- The approach for determining the binding affinities from the NMR data for AP180 binding to clathrin and AP2 α relies on the assumption that exchange between the unbound and bound states is fast and that there are not contributions from intermediate exchange processes on the μs -ms timescale. However, the NMR data presented in Figures 2 and 3 show that some line broadening is observed in the presence of the binding partners, which could be indicative of intermediate exchange. The authors mention CPMG dispersion data (flat curves) but do not show the data to support this claim. These data are critical for validating the assumptions required for estimating the binding affinities by this method.

2. The authors find that binding of the 'extended interaction motif' is not largely affected by point mutations in AP2 $\beta 2$. Given that ~ 70 residues of the AP180 IDR are important for binding to AP2 $\beta 2$, is it expected that the point mutations would have a large effect on the strength of the interaction? Could these interactions be characterized by ITC to confirm that the interaction is minimally perturbed by the AP2 $\beta 2$ mutations?

3. Not all of the Supplementary Figures are referenced in the text.

Reviewer #3 (Remarks to the Author):

Naudi-Fabra et al. study the dynamic interactions between AP2, the adaptor complex for clathrin-mediated endocytosis, and AP180, a clathrin-associated sorting protein. AP180 is large protein with a large intrinsically disordered region (IDR). Its interaction with the adaptor complex was recently discovered but not described in detail. The authors now target the interaction between these two proteins by NMR spectroscopy. They pay particular attention to the IDR of AP180. It is worth mentioning that IDRs are of particular importance for many cellular functions and studying IDRs is therefore particularly important.

The study is thoroughly conducted, and the manuscript is well-written. However, as a non-NMR expert, it is very difficult to follow the results section. It appears to be too specific for a journal with broad readership such as Nature communications. This aspect should be addressed, before considering this manuscript for publication. Please find below some additional comments on the manuscript:

- microMolar, miniMolar etc. should read micromolar (or μM), minimolar (or mM) etc.
- page 6: the affinity constant (K_D) should be termed equilibrium dissociation constant (K_D)
- page 7, 3rd paragraph: "...both top- and side-site may be..." The terms side-site and top-side are confusing. I suggest to change the expression to "...both top and side may be..."
- The cross-linking experiment should be explained in the main text, neither the experiment nor the results are discussed (except two residues which are highlighted on a structure). Conclusions that are drawn should be sufficiently proven. It is not sufficient to provide an upload to a repository (which will most likely not be inspected by the readers)
- The cross-linking results are not provided, this is commonly summarized in form of a table listing all identified cross-links (including sequences, cross-linked residues, scores etc.). In addition, cross-linking results might be visualized in network plots or similar. Number of replicates should also be given (and results for all replicates should be provided in individual tables or a combined overview table).
- Reporting summary: information on the mass spectrometer and the acquisition software are missing
- Methods: Almost all details on the cross-linking experiment(s) are missing, i.e. protein concentrations, cross-linking temperature, experimental details for stage tip clean-up, solvents during LC separation, information on a pre-column (if used), details of the analytical column (length etc.), details on the MS measurements (m/z range, capillary temperature, emitter voltage, analysis mode etc.) as well as database search parameters (peptide length, modifications, charge states etc.) are missing.
- Figure 9 is summarizing the results but is not sufficiently discussed.

We thank the three reviewers for their careful evaluation of our manuscript and for their comments and suggestions for improvement. We have addressed all points raised by the reviewers, which has significantly strengthened our manuscript. Our replies to the reviewers' comments can be found in blue font.

Reviewer #1:

AP180 is a clathrin assembly protein with a very large intrinsically ordered domain (~60 kD) that plays a major role in assembling the complex network of endocytic proteins which underlies clathrin mediated endocytosis. The authors utilized NMR structural mapping to characterize interactions between AP180 and clathrin TD, as well as AP180 and AP2. Indeed they provide strong experimental support for the idea that the intrinsically disordered domain of AP180 contains a large number of clathrin binding sites centered around a DLL/DLF motif. In addition, they also find interactions between clathrin TD and DPF and FxDxF motifs on AP180, motifs previously thought to only be involved in AP2 binding. They go on to characterize interactions between AP180 and the alpha appendage domain of AP2. While indeed they find that AP2 alpha binds to DPF and FxDxF motifs on AP180 as expected, they also find promiscuous binding of AP2 alpha to the clathrin TD binding DLL/DLF motifs of AP180. While these are all weak interactions, something that is believed to be important for the necessary rearrangements to take place as clathrin coated vesicles are forming, they also identified a stronger interaction between AP180 and the beta appendage domain of AP2. While interactions between AP180 and either Clathrin TD or AP2 alpha are in the high micromolar to millimolar range, interactions between AP180 and AP2 beta are in the low micromolar range. This is novel, and informs the mechanism of protein network recruitment to nascent clathrin coated pits.

The strength of this paper lies in their exploitation of NMR spectroscopy to study these interactions in the context of the entire IDP domain of AP180. While some of what they found was expected from previously published work, it is very significant to see these ideas tested experimentally using an intact protein domain. For example, while no binding coupled folding was found in a previous NMR study of clathrin TD binding to a 57aa fragment of AP180 (citation 22 in this paper), some had questioned whether this result would hold up when the entire 60kD IDP domain of AP180 was utilized. Another strength of this paper is that it contributes to both the field of clathrin mediated endocytosis/vesicle trafficking, as well as to our understanding of intrinsically disordered protein domains.

We thank the reviewer for this very positive and enthusiastic assessment of our work.

The study is very well executed, and of extremely high technical proficiency. My only suggestion for improvements are editorial. It would be useful to the reader if the locations of all the binding sites identified are indicated more clearly in the tables. For example, in supplementary table 1, the residue numbers should be indicated at the start and end of each short sequence shown, to allow the reader to better integrate this study with previously published work.

We thank the reviewer for this idea to localize the interaction motifs more precisely within the sequence of AP180. We now include exact numbering of the sequence stretches in Supplementary Table 1 and indicate the surrounding amino acids in Supplementary tables 2 and 3.

Reviewer #2:

In this exciting manuscript from the Milles group, the authors present a tour-de-force characterization of the intrinsically disordered clathrin associated sorting protein AP180 and its interactions with various components of the endocytic machinery. Using solution NMR spectroscopy complemented with isothermal calorimetry and crosslinking mass spectrometry, the authors demonstrate that the long C-terminal IDR of AP180 is disordered in isolation and remains largely disordered and dynamic in complexes with clathrin and the α and $\beta 2$ appendages of the endocytic adaptor complex AP2. A particularly exciting finding of this study is that AP180 utilizes different interaction modes for different partner molecules. AP180 engages clathrin and the AP2 α appendage through a series of weak binding motifs that bind in a somewhat promiscuous manner, whereas a higher affinity interaction between AP180 and the AP2 $\beta 2$ appendage is mediated by a 70 residue 'extended interaction motif'. From these findings, the authors propose a model whereby local concentrations of endocytic proteins at various stages of the endocytic process will bias the AP180 complexes that form and contribute to progression of endocytosis. The results are well-presented, the conclusions are well-supported by the data, and the suggestions included below are minor and are not intended to interfere with publication of this important work.

We thank the reviewer for this highly favorable and encouraging evaluation of our work, as well as for the suggestions to further improve our manuscript, to which we reply below.

1. Several sections of the manuscript could be clarified for a non-NMR audience.

- The authors reference the 'on-off binding rate' in the introduction. Later in the manuscript it becomes clear that this is the k_{ex} value determined from fitting the CPMG relaxation dispersion data. Some explanation of the relationship between k_{ex} and the on and off rate constants would be helpful.

We thank the reviewer for encouraging us to explain the extraction of binding rates using CPMG relaxation dispersion in more detail. Indeed, fitting of the CPMG data will only allow us to extract k_{ex} , which is defined as $k_{on} + k_{off}$ for a two-state exchange reaction. In the case of binding of AP180 to AP2 $\beta 2$, $k_{ex} = k_{on} [AP2\beta 2] + k_{off}$, where $[AP2\beta 2]$ is the concentration of AP2 $\beta 2$. We now explain this more clearly in the revised manuscript.

- SSP values are presented in many figures and mentioned throughout the text, but they are not thoroughly explained. It would be useful to explain how these values can be interpreted. What do the values mean (sign and magnitude)?

We thank the reviewer for pointing this out. SSPs (secondary structure propensities) are derived from carbon secondary chemical shifts, that means the deviation of the measured chemical shift from what is expected for a random coil conformation. They are normalized to between -1 and 1, where -1 would correspond to a fully formed extended conformation and 1 to a fully formed helix. Even though they are displayed per residue, they underlie a window function of 5 residues. We now explain this more clearly in the manuscript.

In general, as also suggested by reviewer 3, we now pay particular attention to thoroughly explain the different NMR parameters and how they can be interpreted to make the manuscript more amenable to a readership without an NMR background.

- The approach for determining the binding affinities from the NMR data for AP180 binding to clathrin and AP2 α relies on the assumption that exchange between the unbound and bound states is fast and that there are not contributions from intermediate exchange processes on the μ s-ms timescale. However, the NMR data presented in Figures 2 and 3 show that some line broadening is observed in the presence of the binding partners, which could be indicative of intermediate exchange. The authors mention CPMG dispersion data (flat curves) but do not show the data to support this claim. These data are critical for validating the assumptions required for estimating the binding affinities by this method.

We thank the reviewer for the suggestion to include (flat) CPMG relaxation dispersion curves for interactions in fast exchange, and we have thus included a new figure (Supplementary Figure 10) demonstrating the absence of intermediate exchange of AP180 upon interaction with the clathrin heavy chain terminal domain (CHC_{TD}). In this Figure (also PbyP Figure 1), we show ΔR_2^{eff} , that means R_2 at the lowest CPMG frequency (31 Hz) minus R_2 at the highest CPMG frequency (1000 Hz), plotted along the sequence of AP180₂₈₁₋₅₀₀ at different concentrations of CHC_{TD}, as well as the CPMG curves of the three residues showing the highest (albeit extremely low) ΔR_2^{eff} . This observation together with chemical shift perturbations upon interaction (as for example shown in Figure 2A) is a signature of fast exchange, at the limit to fast-intermediate exchange for very few residues. We chose to demonstrate flat CPMG curves on this construct, as this showed the largest peak broadening of all AP180 constructs upon interaction with CHC_{TD}.

PbyP Figure 1: CPMG relaxation dispersion of AP180₂₈₁₋₅₀₀ with different concentrations of CHC_{TD}. (A) ΔR_2^{eff} as derived from CPMG relaxation dispersion experiments of 100 μ M ¹⁵N AP180₂₈₁₋₅₀₀ with 40% and 150% CHC_{TD} compared to the concentration of AP180₂₈₁₋₅₀₀. Plotted is the difference between R_2^{eff} at CPMG frequencies of 31.35 Hz and 1000 Hz. (B) CPMG curves of AP180₂₈₁₋₅₀₀ with 40% CHC_{TD} for residues 351Leu, 353Leu and 383Gly, the residues showing the highest ΔR_2^{eff} in this experiment. Data were recorded at a ¹H Larmor frequency of 600 MHz.

The peak broadening observed upon interaction of CHC_{TD} and AP2 α , see Figures 2 and 3, is indeed in line with fast exchange behavior. Peak broadening goes along with increased R_2 (R_{1p}) rates with the full width at half maximum of each peak being proportional to R_2 . R_2 , in turn, depends on the rotational tumbling time of the molecule and, for an intrinsically disordered protein, on its segmental dynamics. We therefore observe small transverse relaxation rates (R_2) for an intrinsically disordered protein (IDP) compared to its molecular weight. When a small region within an IDP binds to a large, folded binding partner, this region will be heavily impacted by the comparatively slow rotational tumbling time of the partner and lead to a local increase in the transverse relaxation rates (shown in Figure 2 and 3, and Supplementary Figures 3-5). This

increase in R_2 ($R_{1\rho}$) translates into peak broadening and is the reason, why peak broadening is observed for AP180 interaction with CHC_{TD} and AP2 α .

In the case of interaction between AP180 and AP2 β , the peak broadening observed around the extended interaction motif (EIM) when only small amounts of AP2 β were bound, appeared to not relate to the $R_{1\rho}$ rates measured for the same interaction. Since we acquired the $R_{1\rho}$ rates with a spin lock field of as high as 1500 Hz, this experiment will effectively quench exchange occurring on the 'CPMG time scale'. Additional broadening would thus be explained by intermediate exchange, which initiated us to perform CPMG relaxation dispersion experiments to assess the interaction between AP180 and AP2 β . This was not the case for interaction with CHC_{TD} and AP2 α . In order to make this reasoning more clear to the readership, we included a more detailed explanation along these lines into the manuscript.

2. The authors find that binding of the 'extended interaction motif' is not largely affected by point mutations in AP2 β . Given that ~ 70 residues of the AP180 IDR are important for binding to AP2 β , is it expected that the point mutations would have a large effect on the strength of the interaction? Could these interactions be characterized by ITC to confirm that the interaction is minimally perturbed by the AP2 β mutations?

Indeed, NMR spin relaxation experiments (Figure 6) suggest that all three tested point mutations of AP2 β do affect binding to AP180. This is demonstrated by the fact that peaks within the EIM remain present also at relatively high concentrations of the AP2 β mutants, albeit significantly broadened, leading to our original conclusion that all presented AP2 β point mutations affect binding, but none inhibits binding.

As the reviewer suggested, we have now performed ITC experiments with AP180₄₀₀₋₆₀₀ and the different AP2 β mutants (PbyP Figure 2). Indeed, compared to the ITC experiments performed with the AP2 β wild type, all mutants weaken the binding to AP180 only mildly, in agreement with our NMR relaxation experiments.

PbyP Figure 2: ITC of AP180₄₀₀₋₆₀₀ with AP2 β 2 mutants. Differential heating powers (DP) per injection are shown on top, enthalpy versus molar ratio of the interaction partners are shown in the bottom row. The data are fitted with a 1:1 binding model resulting in the affinities indicated in the respective graphs (K_D values and binding enthalpies ΔH are shown in the respective graphs). The structure of AP2 β 2 with the different mutations indicated in pink is shown on the right.

Interestingly, while AP2 β 2 K842E and Y888A interact with AP180₄₀₀₋₆₀₀ in an exothermic way, as does the AP2 β 2 wild type, AP2 β 2 Y815A interacts with AP180₄₀₀₋₆₀₀ in an endothermic fashion. Since Y815A is the only mutant in the sandwich domain and both other mutants reside on the platform domain, this suggests that binding to the platform domain may have enthalpic contributions towards binding with AP180, whereas the sandwich domain shows entropic contributions towards binding. Even though our data do not allow to conclude why this might be the case, we find these results so remarkable that we decided to include them into the Supplementary material (Supplementary Figure 13), and we thank the reviewer for suggesting us to perform this experiment.

3. Not all of the Supplementary Figures are referenced in the text.

We thank the reviewer for pointing this out and now included references for all Supplementary Figures in the text.

Reviewer #3:

Naudi-Fabra et al. study the dynamic interactions between AP2, the adaptor complex for clathrin-mediated endocytosis, and AP180, a clathrin-associated sorting protein. AP180 is large protein with a large intrinsically disordered region (IDR). Its interaction with the adaptor complex was recently discovered but not described in detail. The authors now target the interaction between these two proteins by NMR spectroscopy. They pay particular attention to the IDR of AP180. It is worth mentioning that IDRs are of particular importance for many cellular functions and studying IDRs is therefore particularly important.

The study is thoroughly conducted, and the manuscript is well-written. However, as a non-NMR expert, it is very difficult to follow the results section. It appears to be too specific for a journal with broad readership such as Nature communications. This aspect should be addressed, before considering this manuscript for publication. Please find below some additional comments on the manuscript:

We thank the reviewer for this positive assessment of our work and its impact, and for making us aware that our manuscript was in parts hard to read for non NMR experts. We have now thoroughly edited the technical aspects of our manuscript to make it more accessible to non-NMR experts.

For example, we have introduced explanatory sections for the following NMR parameters:

- Secondary chemical shifts (SCS)
- Secondary structure propensities (SSP)
- $R_{1\rho}$ relaxation rates and how they depend on molecular motion
- How NMR peak intensities depend on molecular motion
- What information CPMG relaxation dispersion provides and how to read the data

We acknowledge that the manuscript still contains many technical aspects for a non-NMR expert, but we would like to stress out that the insights gained in this study go far beyond technical insights and have important implications for clathrin mediated endocytosis and complex interaction networks implicating IDPs in a general sense. NMR spectroscopy is the only technique that allows gaining molecular insights into these complex interaction networks and we are thus convinced that

the study merits to report on a thorough technical analysis. We are confident, however, that the current version, comprising more detailed descriptions of the NMR experiments, will make the manuscript accessible for readers from diverse backgrounds.

- microMolar, miniMolar etc. should read micromolar (or μM), minimolar (or mM) etc.

We have applied these changes throughout the manuscript.

- page 6: the affinity constant (K_D) should be termed equilibrium dissociation constant (K_D)

We have adapted this change in the manuscript.

- page 7, 3rd paragraph: "...both top- and side-site may be..." The terms side-site and top-side are confusing. I suggest to change the expression to "...both top and side may be..."

We thank the reviewer for pointing this out. We now changed the description of both binding sites to 'platform' and 'sandwich' domains, which should be less confusing.

- The cross-linking experiment should be explained in the main text, neither the experiment nor the results are discussed (except two residues which are highlighted on a structure). Conclusions that are drawn should be sufficiently proven. It is not sufficient to provide an upload to a repository (which will most likely not be inspected by the readers)

We thank the reviewer of pointing this out. We have now included a more detailed description of the mass spectrometry (MS) experiments and results and we have also included them in more detail into the discussion. In addition to the data upload, we now provide a list of all peptides detected by MS including the cross-links observed.

- The cross-linking results are not provided, this is commonly summarized in form of a table listing all identified cross-links (including sequences, cross-linked residues, scores etc.). In addition, cross-linking results might be visualized in network plots or similar. Number of replicates should also be given (and results for all replicates should be provided in individual tables or a combined overview table).

We thank the reviewer for making us aware of this shortcoming. We have now included a list of all cross-linked peptides, their mass and modifications, together with a short description of the cross-linked peptides/residues. This list contains all unique peptides, while the information on all detected spectra (comprising also multiple detections of the same peptide) appear on the data upload in ProteomXchange. This upload also contains the scores for every individual detection.

While AP2 β 2 comprises 15 lysines, AP180₄₃₀₋₅₀₀ contains only one single lysine, close to its C-terminus. This single lysine in AP180₄₃₀₋₅₀₀ cross-links with two lysines within AP2 β 2 that are in close spatial proximity, but no other lysines within the whole AP2 β 2 appendage domain. The close spatial proximity of the two cross-linked lysines within AP2 β 2 gives us confidence in the obtained data, such that only one replicate was performed.

We would also like to stress that NMR chemical shift titrations of the ^{15}N labelled AP2 β 2 upon interaction with unlabelled AP180₄₃₀₋₅₀₀ provide detailed insights into the binding surface of AP2 β 2 for AP180₄₃₀₋₅₀₀, which is in very good agreement with the cross-linking MS data between AP180₄₃₀₋

⁵⁰⁰ and AP2β2. Together, NMR and MS thus provide a very robust description of the interaction surfaces between the two partners. We now point this out more clearly in the revised manuscript.

Since AP180 likely engages in a dynamic binding mode onto the surface of the AP2β2 appendage domain, we refrain from visualizing those two cross-links using a single binding pose. However, we now visualize the position of the single lysine contained in AP180₄₃₀₋₅₀₀ (see PbyP Figure 3), which we also include as a new sub-panel in Figure 7 (Figure 7D).

PbyP Figure 3: Cross-linking network between AP180₄₃₀₋₅₀₀ and AP2β2. The PDB structure of AP2β2 (1E42) is shown in gray and one conformation of AP180₄₃₀₋₅₀₀ is shown in blue. All lysines in both proteins are illustrated as balls. Those not involved in inter-molecular cross-links are colored pink. Lysines involved in inter-molecular cross-links between AP180₄₃₀₋₅₀₀ and AP2β2 are shown in orange. Cross-links are illustrated as orange lines. Residues showing dispersion in a CPMG relaxation dispersion experiment (see Figure 4E) are colored in light orange on AP180 and illustrate the residues in contact with AP2β2. Note that the organization of the two proteins in the figure does not reflect an actual binding mode, and only serves the purpose of illustrating inter-molecular cross-links detected by mass spectrometry.

- Reporting summary: information on the mass spectrometer and the acquisition software are missing

We thank the reviewer for pointing out this missing information. We have included the instrument type (Orbitrap Exploris 480) and the acquisition software (Thermo Scientific Xcalibur 4.5.474.0) into the reporting summary.

- Methods: Almost all details on the cross-linking experiment(s) are missing, i.e. protein concentrations, cross-linking temperature, experimental details for stage tip clean-up, solvents during LC separation, information on a pre-column (if used), details of the analytical column (length etc.), details on the MS measurements (m/z range, capillary temperature, emitter voltage, analysis mode etc.) as well as database search parameters (peptide length, modifications, charge states etc.) are missing.

We have included the information the reviewer suggested into the methods section of the revised manuscript.

- Figure 9 is summarizing the results but is not sufficiently discussed.

We thank the reviewer for encouraging us to discuss our model in more detail. We have thus included a more detailed discussion of Figure 9 in accordance with our data.

REVIEWERS' COMMENTS

Reviewer #2 (Remarks to the Author):

The authors have done an outstanding job of addressing all of my prior concerns. I have no further suggestions and hope that this manuscript can advance to publication without further delay.

Reviewer #3 (Remarks to the Author):

The authors have taken into account all my comments. I cannot find the cross-linking data in the repository but assume that this problem will be solved upon acceptance. I have no further comments.